# DOVE: Efficient One-Step Diffusion Model for Real-World Video Super-Resolution

**Zheng Chen**[1]\*, **Zichen Zou**[2]\*, **Kewei Zhang**[1], **Xiongfei Su**[3],
**Xin Yuan**[4], **Yong Guo**[5], **Yulun Zhang**[1]†

[1]School of Computer Science, Shanghai Jiao Tong University,
[2]Zhiyuan College, Shanghai Jiao Tong University, [3]China Mobile Research Institute,
[4]Westlake University, [5]Huawei Consumer Business Group

## Abstract

Diffusion models have demonstrated promising performance in real-world video super-resolution (VSR). However, the dozens of sampling steps they require, make inference extremely slow. Sampling acceleration techniques, particularly single-step, provide a potential solution. Nonetheless, achieving one step in VSR remains challenging, due to the high training overhead on video data and stringent fidelity demands. To tackle the above issues, we propose DOVE, an efficient one-step diffusion model for real-world VSR. DOVE is obtained by fine-tuning a pretrained video diffusion model (*i.e.*, CogVideoX). To effectively train DOVE, we introduce the latent-pixel training strategy. The strategy employs a two-stage scheme to gradually adapt the model to the video super-resolution task. Meanwhile, we design a video processing pipeline to construct a high-quality dataset tailored for VSR, termed HQ-VSR. Fine-tuning on this dataset further enhances the restoration capability of DOVE. Extensive experiments show that DOVE exhibits comparable or superior performance to multi-step diffusion-based VSR methods. It also offers outstanding inference efficiency, achieving up to a **28×** speed-up over existing methods such as MGLD-VSR. Code is available at: `https://github.com/zhengchen1999/DOVE`.

## 1 Introduction

Video super-resolution (VSR) is a long-standing task that aims to reconstruct high-resolution (HR) videos from low-resolution (LR) inputs [11, 16]. With the rapid growth of smartphone photography and streaming media, real-world VSR has become increasingly critical. Unlike the synthetic degradations (*e.g.*, bicubic), real-world videos often suffer from complex and unknown degradation. This makes high-quality video restoration difficult. To tackle this issue, numerous methods have been proposed [51, 26, 37, 47, 60]. Among them, generative models, *e.g.*, generative adversarial networks (GANs), are widely adopted for their ability to synthesize fine details [8, 23, 5].

Recently, a new generative model, the diffusion model (DM), has rapidly gained popularity [10]. Compared with GANs, diffusion models exhibit stronger generative capabilities, especially those pretrained on large-scale datasets [28, 27, 2, 59, 52]. Therefore, leveraging pretrained diffusion models for VSR has become an increasingly popular direction. For instance, some methods adopt pretrained text-to-image (T2I) models [50, 63] and incorporate temporal layers and optical flow constraints to ensure consistency across frames. Meanwhile, some approaches directly employ the text-to-video (T2V) model [9, 48], and use ControlNet to constrain video generation. By exploiting the natural priors in pretrained models, these methods can realize more realistic restorations.

---

\*Equal contribution.
†Corresponding author: Yulun Zhang, yulun100@gmail.com

39th Conference on Neural Information Processing Systems (NeurIPS 2025).

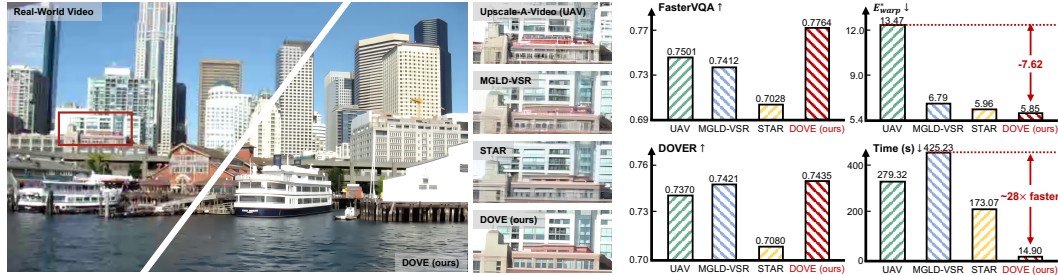

Figure 1: Efficiency and performance comparisons on the real-world benchmark (*i.e.*, VideoLQ [5]). We provide qualitative (left) and quantitative (right) results. The running time (Time) is measured on one A100 GPU using a 33-frame 720×1280 video. Our method achieves impressive performance and excellent efficiency. Compared with MGLD-VSR [50], DOVE is approximately **28×** faster.

However, existing diffusion-based VSR methods face several critical challenges: **(1) Multi-step sampling restricts efficiency.** To generate high-quality HR videos, these models typically require dozens of sampling steps. This seriously hinders the running efficiency. Moreover, for some methods [50, 48], long videos are processed in time segments, which further amplifies this inefficiency. **(2) Additional modules increase overhead.** Whether based on T2I or T2V models, many methods often introduce auxiliary components [63, 9], *e.g.*, ControlNet [57] or temporal layers, to realize VSR. These additional modules further slow down inference. For example, when processing a 33-frame 720p (720×1280) video on an NVIDIA A100 GPU, MGLD-VSR [50] takes 425.23 seconds, while STAR [48] requires 173.07 seconds. Such high latency severely hinders the application of diffusion-based video super-resolution methods in the real world.

One common approach to accelerating the diffusion model is to reduce the number of inference steps [33, 22]. In this context, single-step inference has attracted widespread attention as an extreme acceleration form. Prior studies have demonstrated that single-step diffusion models achieve impressive results in image/video generation [30, 54, 61, 18] and image restoration [39, 45, 13] (*e.g.*, image super-resolution) tasks. However, in VSR, single-step diffusion models have rarely been studied. There are two critical difficulties in realizing one-step inference in VSR: **(1) Excessive video-training cost.** To enhance single-step models performance, some methods (*e.g.*, DMD [55] and VSD [41, 45]) jointly optimize multiple networks. While manageable in the image domain, such overhead becomes burdensome and unacceptable in video due to the multi-frame setting. **(2) High-fidelity demands in VSR.** Some single-step video generation models improve generation quality via adversarial training rather than multi-network distillation [61, 18]. However, the inherent instability of adversarial training may introduce undesired details in results, hindering VSR performance [45].

To address these challenges, we propose DOVE, an efficient one-step diffusion model for real-world video super-resolution. It is built by fine-tuning an advanced pretrained video generation model (*i.e.*, CogVideoX [52]). Considering the strong representation and prior knowledge of the pretrained model, we do not introduce additional components (*e.g.*, optical flow module [50, 63] or ControlNet [48]). This design can further improve the inference efficiency.

For effective DOVE training, we introduce the latent-pixel training strategy. Based on the above analysis, we opt for the regression loss instead of distillation or adversarial losses to enhance training efficiency. The strategy consists of two stages: **Stage-1: Adaptation.** In the latent space, we minimize the gap between the predicted and HR latent representation. This enables the model to learn one-step LR-to-HR mapping. **Stage-2: Refinement.** In the pixel space, we perform mixed training using both images and short video clips. This stage enhances model restoration performance. With the proposed training strategy, we can complete model fine-tuning within only 10K iterations.

Moreover, fine-tuning pretrained models on high-quality datasets is crucial for achieving strong performance. However, in the field of VSR, suitable public datasets [63, 25] remain scarce. To address this issue, we design a systematic video processing pipeline. Using the pipeling, we curate a high-quality dataset of 2,055 videos, HQ-VSR, from existing large-scale sources. Equipped with the latent-pixel training strategy and the HQ-VSR dataset, our DOVE achieves impressive performance.

As shown in Fig. 1, DOVE outperforms state-of-the-art multi-step diffusion-based VSR methods. Simultaneously, due to one-step inference, our DOVE is up to **28×** faster over previous diffusion-based VSR methods, *e.g.*, MGLD-VSR [50]. In summary, our contributions include:

- We propose a novel one-step diffusion model, DOVE, for real-world VSR. To our knowledge, this is the first diffusion-based VSR model with one-step inference.
- We design a latent-pixel training strategy and develop a video processing pipeline to construct a high-quality dataset tailored for VSR, enabling effective fine-tuning of DOVE.
- Extensive experiments demonstrate that DOVE achieves state-of-the-art performance across multiple benchmarks with remarkable efficiency.

## 2 Related Work

### 2.1 Video Super-Resolution

With the advancement of deep learning, numerous video super-resolution (VSR) methods have emerged [11, 24, 16, 4]. These approaches (*e.g.*, BasicVSR [4] and Vrt [16]) utilize a variety of architectures, including recurrent-based [17, 32] and sliding-window-based [14, 53] models, and have demonstrated promising results. However, these methods typically assume a fixed degradation process [49, 53], which limits their performance when confronted with real-world degradation, which is often more complex. To better address such scenarios, some methods, such as RealVSR [51] and MVSR4x [37], have utilized HR-LR paired data from real environments. In contrast, others (*e.g.*, RealBasicVSR [5]) have proposed variable degradation pipelines to enhance the model's adaptability to the complex degradations in real-world VSR. In addition, real-world VSR methods [60, 26, 47] incorporate structural modifications to tackle these challenges. Despite the considerable development in these domains, these methods still face persistent challenges in generating delicate textures.

### 2.2 Diffusion Model

Diffusion models [10] have demonstrated strong performance in visual tasks, driving the development of both image generation models [28, 27, 29] and video generation models [2, 59]. These pretrained models (*e.g.*, Stable Diffusion [29] and I2VGen-XL [59]) offer rich generative priors, significantly advancing downstream tasks such as video restoration. By leveraging pretrained diffusion models, many video restoration methods [50, 63, 9, 48, 36, 15] can recover more realistic video textures. Some approaches [50, 63, 15] adapt pretrained image generation models for VSR tasks, enhancing them to address temporal inconsistencies between frames. For example, Upscale-A-Video [63] employs temporal layers to train on a frozen pretrained model, thereby enhancing the temporal consistency of the video. Meanwhile, other approaches [9, 48] utilize video generation models [59] and incorporate ControlNet [57] to constrain video generation. However, these methods remain limited by multi-step sampling, which hampers efficiency, and by the added modules, which increase computational overhead, thereby impacting inference speeds.

### 2.3 One-Step Acceleration

A common method to accelerate diffusion models is reducing the number of inference steps, with one-step acceleration gaining significant attention as an extreme approach. This technique leverages methods such as rectified flow [19, 20], adversarial training [18, 61], and score distillation [55, 41]. Building on these methods, recent image super-resolution (ISR) studies [40, 13, 45, 7] have integrated one-step diffusion. For instance, OSEDiff [45] applies variational score distillation [41] to improve inference efficiency. However, directly applying these methods to VSR is impractical due to the high computational cost associated with processing multiple video frames. Additionally, some methods (*e.g.*, SF-V [61] and Adversarial Post-Training [18]) have employed adversarial training to explore one-step diffusion in video generation. However, adversarial training methods are inherently unstable. Applying them in VSR, instead of multi-network distillation [55, 41], may introduce unwanted artifacts that degrade the VSR performance [45]. In this paper, we introduce a novel and effective one-step diffusion model that successfully accelerates the inference process of video super-resolution.

## 3 Method

In this section, we introduce the proposed efficient one-step diffusion model, DOVE. First, we present the overall framework of DOVE, which is developed based on CogVideoX to achieve one-step video super-resolution (VSR). Then, we describe two key designs that facilitate high-quality fine-tuning: the latent-pixel training strategy and the video processing pipeline.

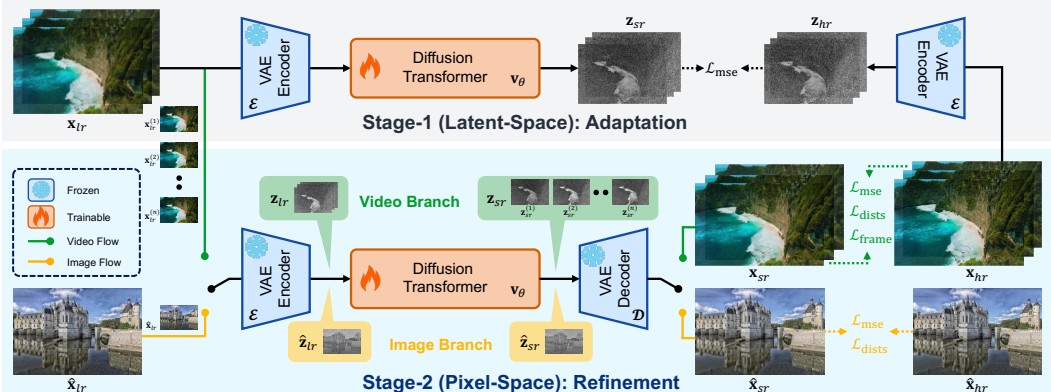

Figure 2: Overview of the framework and training strategy of DOVE. Our method performs one-step sampling to reconstruct HR videos ($\mathbf{x}_{sr}$) from LR inputs ($\mathbf{x}_{lr}$). To enable effective training, we adopt the two-stage latent-pixel training strategy. **Stage-1 (latent-space):** Minimize the difference between the predicted and HR latents. **Stage-2 (pixel-space):** Improve detail generation using mixed **image** / **video** training, where the data branch at each iteration is controlled by image ratio ($\varphi$). To reduce memory cost, video is processed frame-by-frame in the encoder and decoder.

## 3.1 Overall Framework

We construct our one-step diffusion network, DOVE, based on a powerful pretrained text-to-video model (*i.e.*, CogVideoX [52]). CogVideoX employs a 3D causal VAE to compress videos into the latent space and uses a Transformer denoiser $\mathbf{v}_\theta$ for diffusion. Leveraging its strong priors, our method can better handle the complexities of real-world scenarios. Meanwhile, to enhance inference efficiency, we do not introduce any auxiliary modules like previous methods [50, 63, 9], such as temporal layers or ControlNet. Instead, we design a two-stage training strategy and curate a high-quality dataset specifically for the VSR task, enabling strong performance with high efficiency.

The overall architecture of DOVE is illustrated in Fig. 2. Specifically, given a low-resolution (LR) video $\mathbf{x}_{lr}$, we first upscale it to the target high resolution using bilinear interpolation. The upscaled video is then encoded into a latent representation $\mathbf{z}_{lr}$ by the VAE encoder $\mathcal{E}$. Following previous works [46, 45], we take $\mathbf{z}_{lr}$ as the starting point of the diffusion process. That is, we treat $\mathbf{z}_{lr}$ as the noised latent $\mathbf{z}_t$ at a specific timestep $t$, which is originally formulated as:

$$\mathbf{z}_t = \sqrt{\bar{\alpha}_t}\mathbf{z} + \sqrt{1 - \bar{\alpha}_t}\boldsymbol{\epsilon}, \quad \bar{\alpha}_t = \prod_{s=1}^{t}(\alpha_t), \quad \alpha_t = 1 - \beta_s, \tag{1}$$

where $\boldsymbol{\epsilon}$ is the Gaussian noise, $\mathbf{z}$ is the "clean" latent sample, and $\beta$ is the noise factor. Afterwards, a single denoising step is performed through Transformer $\mathbf{v}_\theta$, yielding the "clean" latent $\mathbf{z}_{sr}$ (*i.e.*, $\mathbf{z}$ in Eq. (1)). Since CogVideoX adopts the $v$-prediction formulation, the denoising process is defined as:

$$\mathbf{z}_{sr} = \sqrt{\bar{\alpha}_t}\mathbf{z}_{lr} - \sqrt{1 - \bar{\alpha}_t}\mathbf{v}_\theta\left(\mathbf{z}_{lr}, \mathbf{c}, t\right). \tag{2}$$

Unlike the previous approach [45] that set $t$ to the total diffusion step (*i.e.*, 999 in CogVideoX), we choose a smaller value. This is based on the observation that early diffusion steps focus on global structure while later steps refine fine details. Since the LR video already contains sufficient structural information, starting from the beginning is not needed. Conversely, a tiny $t$ (*i.e.*, late step) would hinder the removal of degradation. Therefore, we empirically set $t$=399. Finally, the latent $\mathbf{z}_{sr}$ is decoded by the VAE decoder $\mathcal{D}$ to obtain the output video $\mathbf{x}_{sr}$, as the restoration results.

## 3.2 Latent-Pixel Training Strategy

The framework described above enables us to adapt a generative model for the VSR task. However, since the model is designed for multi-step sampling and the distribution of $\mathbf{z}_{lr}$ and $\mathbf{z}_t$ are not consistent, the pretrained model cannot be directly applied. Thus, fine-tuning is required to ensure that the reconstructed output $\mathbf{x}_{sr}$, closely matches the high-resolution (HR) ground truth $\mathbf{x}_{hr}$.

Nevertheless, many one-step fine-tuning strategies developed for images (*e.g.*, DMD [55] and VSD [41] are not available in the video domain, as the video data volume is larger (due to multiple frames). Besides, adversarial training, which is commonly applied in single-step video generation [61, 18], is less suitable for VSR. This is because the inherent instability of adversarial training may lead to undesired details in the results, which are misaligned with the high-fidelity requirements of VSR.

To enable effective training for DOVE, we design a novel latent-pixel training strategy. We adopt the regression loss, instead of distillation or adversarial learning, to improve training efficiency. In addition, we only fine-tune the Transformer component, preserving the priors in the pretrained VAE. The strategy is illustrated in Fig. 2, which consists of a two-stage training process.

**Stage-1: Adaptation.** With the VAE decoder $\mathcal{D}$ fix, making the predicted $\mathbf{x}_{sr}$ close to the high-quality $\mathbf{x}_{hr}$, corresponds to minimizing the difference between $\mathbf{z}_{sr}$ and the HR video latent $\mathbf{z}_{hr}$.

Due to the high compression ratio of the VAE, training in latent space is more efficient than in pixel space. Therefore, we first minimize the difference between $\mathbf{z}_{sr}$ and $\mathbf{z}_{hr}$. As illustrated in Fig. 2, both the LR and HR videos are first encoded into latent representations via the VAE encoder $\mathcal{E}$. We then train the Transformer $\mathbf{v}_{\theta}$ using the MSE loss (denoted as $\mathcal{L}_{s1}$). The process is denoted as:

$$\mathcal{L}_{s1} = \mathcal{L}_{\text{mse}}(\mathbf{z}_{sr}, \mathbf{z}_{hr}) = \frac{1}{|\mathbf{z}_{hr}|} \left\| \mathbf{z}_{sr} - \mathbf{z}_{hr} \right\|_2^2. \tag{3}$$

Benefiting from the high computational efficiency in the latent space, we can train the model on videos with longer frame sequences. This enables the model to better handle long-duration video inputs, which are common but challenging in video super-resolution tasks.

**Stage-2: Refinement.** After the first training stage, the model can achieve LR to HR video mapping to a certain extent. However, we observe that the output $\mathbf{x}_{sr}$ exhibits noticeable gaps from the ground-truth $\mathbf{x}_{hr}$. This may be because in the latent space, although $\mathbf{z}_{sr}$ is close to $\mathbf{z}_{hr}$, the slight gap will be further amplified after passing through the VAE decoder $\mathcal{D}$. Therefore, further fine-tuning in pixel space is necessary to reduce reconstruction errors. However, the large volume of video data makes the training overhead in pixel space unacceptable.

To address this issue, we introduce image data for the second stage fine-tuning. An image can be treated as a single-frame video, whose data volume is much smaller than that of multi-frame sequences, making pixel-domain training feasible. As illustrated in Fig. 2, we minimize the MSE loss between the output image $\hat{\mathbf{x}}_{sr}$ and the HR image $\hat{\mathbf{x}}_{hr}$. Meanwhile, we additionally adopt the perceptual DISTS [6] loss, $\mathcal{L}_{\text{dists}}$, to better preserve texture details. The total image loss is defined as:

$$\mathcal{L}_{\text{s2-image}} = \mathcal{L}_{\text{mse}}(\hat{\mathbf{x}}_{sr}, \hat{\mathbf{x}}_{hr}) + \lambda_1 \, \mathcal{L}_{\text{dists}}(\hat{\mathbf{x}}_{sr}, \hat{\mathbf{x}}_{hr}), \tag{4}$$

where $\lambda_1$ is the perceptual loss scaler. After fine-tuning on images in pixel space, the restoration detail is greatly improved. However, since the model is trained only on single-frame data, it exhibits instability when handling multi-frame videos, which adversely affects overall performance. To resolve this issue, we reintroduce video data and adopt a mixed image/video training strategy.

Specifically, we revisited the memory bottlenecks during video training. We find that the VAE is the primary constraint. Inspired by the success of image-only training, we process videos frame-by-frame through the VAE encoder $\mathcal{E}$ and decoder $\mathcal{D}$. This avoids multi-frame memory spikes. Meanwhile, the Transformer continues to operate on the complete latent. The process (as in Fig. 2) is defined as:

$$\mathbf{z}_{lr}^{(t)} = \mathcal{E}(\mathbf{x}_{lr}^{(t)}), \ t = 1, \ldots, n, \quad \mathbf{z}_{lr} = \left[\mathbf{z}_{lr}^{(1)}, \mathbf{z}_{lr}^{(2)}, \ldots, \mathbf{z}_{lr}^{(n)}\right], \quad \mathbf{z}_{sr} = \Phi_{\theta}(\mathbf{z}_{lr}),$$
$$\mathbf{x}_{sr}^{(t)} = \mathcal{D}(\mathbf{z}_{sr}^{(t)}), \ t = 1, \ldots, n, \quad \mathbf{x}_{sr} = \left[\mathbf{x}_{sr}^{(1)}, \mathbf{x}_{sr}^{(2)}, \ldots, \mathbf{x}_{sr}^{(n)}\right], \tag{5}$$

where $\mathbf{x}_{lr}^{(t)}$ and $\mathbf{x}_{sr}^{(t)}$ mean the $t$-th frame of LR and SR video, respectively; $n$ is the frame number; $\Phi_{\theta}(\cdot)$ denotes the one-step denoising process based on Transformer $\mathbf{v}_{\theta}$ defined in Eq. (2). As with the image-level training, we apply MSE loss and the perceptual DISTS loss. Additionally, to enforce frame consistency, we introduce a frame difference loss:

$$\mathcal{L}_{\text{frame}}(\mathbf{x}_{sr}, \mathbf{x}_{hr}) = \frac{1}{n-1} \sum_{t=2}^{n} \left\| \Delta \mathbf{x}_{\text{sr}}^{(t)} - \Delta \mathbf{x}_{\text{hr}}^{(t)} \right\|_1, \quad \Delta \mathbf{x}^{(t)} := \mathbf{x}^{(t)} - \mathbf{x}^{(t-1)}, \tag{6}$$

where $\Delta \mathbf{x}^{(t)}$ captures the frame-to-frame change. The total loss for video training is computed as:

$$\mathcal{L}_{\text{s2-video}} = \mathcal{L}_{\text{mse}}(\mathbf{x}_{sr}, \mathbf{x}_{hr}) + \lambda_1 \, \mathcal{L}_{\text{dists}}(\mathbf{x}_{sr}, \mathbf{x}_{hr}) + \lambda_2 \, \mathcal{L}_{\text{frame}}(\mathbf{x}_{sr}, \mathbf{x}_{hr}), \tag{7}$$

where $\lambda_2$ is the frame loss scaler. By including video data in training, the stability of video processing is improved, further enhancing video restoration performance.

In summary, our mixed training strategy (combining image and video) in the pixel domain, effectively enhances restoration performance and robustness on videos. Besides, we introduce a hyperparameter $\varphi$ to control the ratio between image and video samples. We study the effect of $\varphi$ in Sec. 4.2.

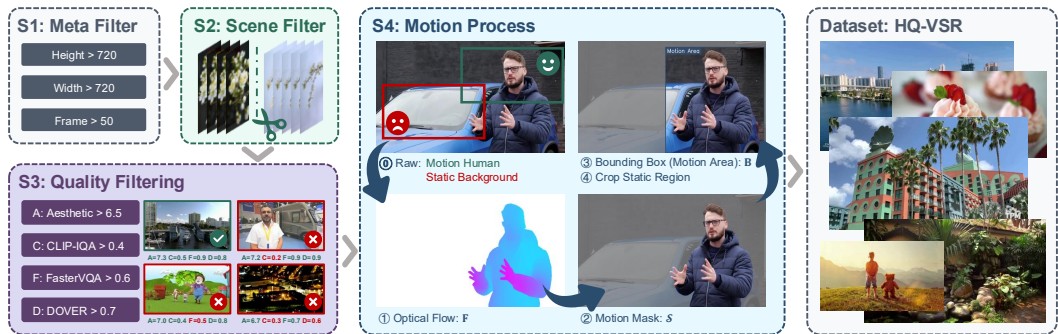

Figure 3: The illustration of the video processing pipeline (four steps). Based on this pipeline, we construct HQ-VSR, a high-quality dataset tailored for the VSR task.

## 3.3 Video Processing Pipeline

**Motivation.** Fine-tuning on high-quality datasets can significantly improve the performance of pretrained models on downstream tasks. However, in the domain of video super-resolution (VSR), suitable public datasets remain scarce. Current VSR methods typically apply the following data: **(1)** conventional video datasets, *e.g.*, REDS [24]; **(2)** self-collected videos, *e.g.*, YouHQ [63]; and **(3)** publicly text-video datasets, *e.g.*, WebVid-10M [1] and OpenVid-1M [25].

Nevertheless, these datasets have some limitations: **(1)** some contain relatively few data and scenarios; **(2)** some lack proper curation or filtering specifically for VSR. As a result, fine-tuning on these datasets cannot fully realize the potential of pretrained models in the VSR task.

**Pipeline.** To address the aforementioned limitations, we propose a systematic video processing pipeline to construct a high-quality dataset tailored for VSR. As shown in Fig. 3, the pipeline is:

*Step 1: Metadata Filtering.* We begin with a coarse filtering process based on metadata, *i.e.*, video resolution and frame count. We extract all videos with a shorter side that exceeds 720 pixels and over 50 frames. As VSR often targets large-sized videos, the matching training data is required.

*Step 2: Scene Filtering.* Following prior work [2], we perform scene detection, segmenting videos into distinct scenes. We discard short clips with fewer than 50 frames. This step reduces cuts and transitions, which are unsuitable for the model to learn coherent video semantics.

*Step 3: Quality Filtering.* Next, we score each video with multiple quality metrics. While previous works [62, 25] have used metrics like the LAION aesthetic model [31], these are insufficient, since VSR pays more attention to detail quality. Therefore, we incorporate more metrics: CLIP-IQA [35], FasterVQA [43], and DOVER [44]. With more metrics, we achieve stricter filtering.

*Step 4: Motion Processing.* Finally, we eliminate videos with insufficient motion. We first estimate optical flow to calculate the motion score, following prior methods [62, 2]. Although the score can filter some still videos, the global score is suboptimal for VSR. In VSR, training data is typically generated via cropping rather than resizing from HR video to preserve fine details. However, high-motion videos may contain static regions (*e.g.*, speech backgrounds in Fig. 3), yielding static crops.

To address this, we introduce the motion area detection algorithm for localized processing (see Fig. 3). We generate a motion intensity map $M$ from the optical flow $\mathbf{F}$, and apply a threshold $\tau$ to produce a motion mask. Then, we determine the motion areas based on the mask. To ensure sufficient context, we expand the bounding box by a fixed padding $p$. The procedure is defined as:

$$M_{ij} = \|\mathbf{F}_{ij}\|_2, \quad \mathcal{S} = \{\,(i,j) \in \Omega \mid M_{ij} > \tau\},$$
$$\mathbf{B} = \big(\min_{(i,j)\in\mathcal{S}} i - p, \ \min_{(i,j)\in\mathcal{S}} j - p, \ \max_{(i,j)\in\mathcal{S}} i + p, \ \max_{(i,j)\in\mathcal{S}} j + p\big), \tag{8}$$

where $\Omega \subset \mathbb{Z}^2$ is the set of all pixel indices; $\mathbf{F}_{ij}$ denotes the optical-flow vector at pixel $(i, j)$; $\mathcal{S}$ is the motion mask; and $\mathbf{B}$ is the bounding box corresponding to motion areas. Finally, we crop the video according to the bounding box $\mathbf{B}$, and discard the cropped region with resolution lower than 720p.

**HQ-VSR.** We apply the proposed pipeline to the public dataset OpenVid-1M [25], which contains diverse scenes. Based on this, we extract 2,055 high-quality video samples suitable for VSR, forming a new dataset, HQ-VSR. The detailed pipeline configuration is provided in the supplementary material. Fine-tuning our DOVE on the HQ-VSR yields superior performance compared to other datasets.

| Training Stage | S1 | S1+S2-I | S1+S2-I/V |
|---|---|---|---|
| PSNR ↑ | **27.20** | 26.39 | 26.48 |
| LPIPS ↓ | 0.3037 | 0.2784 | **0.2696** |
| CLIP-IQA ↑ | 0.3236 | 0.5085 | **0.5107** |
| DOVER ↑ | 0.6154 | 0.7694 | **0.7809** |

(a) Ablation on training strategy.

| Image Ratio | 0% (video) | 20% | 50% | 80% | 100% (image) |
|---|---|---|---|---|---|
| PSNR ↑ | 26.41 | 26.41 | 26.44 | **26.48** | 26.39 |
| LPIPS ↓ | 0.2624 | **0.2617** | 0.2686 | 0.2696 | 0.2784 |
| CLIP-IQA ↑ | 0.4800 | 0.5012 | 0.5027 | **0.5107** | 0.5085 |
| DOVER ↑ | 0.7647 | 0.7701 | 0.7751 | **0.7809** | 0.7694 |

(b) Ablation on image ratio ($\varphi$) in stage-2.

| Dataset | PSNR ↑ | LPIPS ↓ | CLIP-IQA ↑ | DOVER ↑ |
|---|---|---|---|---|
| YouHQ | 26.88 | 0.3383 | 0.2496 | 0.3965 |
| OpenVid-1M | 27.04 | 0.3376 | 0.2683 | 0.4363 |
| HQ-VSR | **27.20** | **0.3037** | **0.3236** | **0.6154** |

(c) Ablation on training dataset.

| Pipeline | PSNR ↑ | LPIPS ↓ | CLIP-IQA ↑ | DOVER ↑ |
|---|---|---|---|---|
| OpenVid-1M | 27.04 | 0.3376 | 0.2683 | 0.4363 |
| +Filter | 27.09 | 0.3236 | 0.2894 | 0.5357 |
| +Motion | **27.20** | **0.3037** | **0.3236** | **0.6154** |

(d) Ablation on processing pipeline.

Table 1: Ablation study. Evaluation is conducted on UDM10. (a) S1/S2: stage-1/2; I: image-only training; I/V: image-video mixed training. (b) 0%: video-only; 100%: image-only. (c): Experiments on stage-1. (d) +Filter: apply steps 1~3; +Motion: further apply step 4 (motion processing).

# 4 Experiments

## 4.1 Experimental Settings

**Datasets.** The training dataset comprises video and image datasets. The video dataset, HQ-VSR, includes 2,055 high-quality videos, and adopts the RealBasicVSR [5] degradation pipeline to synthesize LQ-HQ pairs. The image dataset is DIV2K [3], with 900 images, which follows the Real-ESRGAN [38] degradation process. For evaluation, we apply both synthetic and real-world datasets. The synthetic datasets include UDM10 [34], SPMCS [53], and YouHQ40 [63], using the same degradations as training. For real-world datasets, we apply RealVSR [51], MVSR4x [37], and VideoLQ [5]. RealVSR and MVSR4x contain real-world LQ-HQ pairs captured via mobile phones, while VideoLQ is Internet-sourced without HQ references. All experiments are conducted with a scaling factor $\times 4$.

**Evaluation Metrics.** We adopt multiple evaluation metrics to assess model performance, which are categorized into two types: image quality assessment (IQA) and video quality assessment (VQA). The IQA metrics include two fidelity measures: PSNR and SSIM [42]. We also use some perceptual quality IQA metrics: LPIPS [58], DISTS [6], and CLIP-IQA [35]. For VQA, we employ FasterVQA [43] and DOVER [44] to evaluate overall video quality. Meanwhile, we adopt the flow warping error $E^*_{warp}$, refers to $E_{warp}$ ($\times 10^{-3}$) [12], to assess temporal consistency. Through these metrics, we conduct a comprehensive evaluation of video quality.

**Implementation Details.** Our DOVE is based on the text-to-video model, CogVideoX1.5 [52]. We use an empty text as the prompt, which is pre-encoded in advance to reduce inference overhead. The proposed two-stage training strategy is then applied for fine-tuning. Both stages are trained on 4 NVIDIA A800-80G GPUs with the total batch size 8. We use the AdamW optimizer [21] with $\beta_1$=0.9, $\beta_2$=0.95, and $\beta_3$=0.98. In stage-1, training is conducted on video data. The videos have a resolution of $320 \times 640$ and a frame length of 25. The model is trained for 10,000 iterations with a learning rate of $2 \times 10^{-5}$. In stage-2, both video and image data are used, with $\varphi$=0.8 (*i.e.*, images comprising 80% of the input). All inputs have a resolution of $320 \times 640$. The model is trained for 500 iterations with a learning rate of $5 \times 10^{-6}$. The loss weights $\lambda_1$ and $\lambda_2$ are set to 1.

## 4.2 Ablation Study

We investigate the effectiveness of the proposed latent-pixel training strategy and video processing pipeline. All training configurations are kept consistent with settings described in Sec. 4.1. We evaluate all models on UDM10 [34]. Results are presented in Tab. 1.

**Training Strategy.** We study the effects of the latent-pixel training strategy, as shown in Tab. 1a. In the stage-1 (S1), where training is conducted in latent space with MSE loss, the results tend to be overly smooth, leading to lower perceptual performance. After fine-tuning in pixel space during stage-2 (S2), perceptual metrics (*i.e.*, LPIPS, CLIP-IQA, and DOVER), improve significantly. Furthermore, using a mixed training scheme with both images and videos in stage-2 (S2-I/V) leads to further performance gains, demonstrating the effectiveness of hybrid training.

| Dataset | Metric | RealESRGAN [38] | ResShift [56] | RealBasicVSR [5] | Upscale-A-Video [63] | MGLD-VSR [50] | VEnhancer [9] | STAR [48] | DOVE (ours) |
|---|---|---|---|---|---|---|---|---|---|
| UDM10 | PSNR ↑ | 24.04 | 23.65 | 24.13 | 21.72 | 24.23 | 21.32 | 23.47 | 26.48 |
| | SSIM ↑ | 0.7107 | 0.6016 | 0.6801 | 0.5913 | 0.6957 | 0.6811 | 0.6804 | 0.7827 |
| | LPIPS ↓ | 0.3877 | 0.5537 | 0.3908 | 0.4116 | 0.3272 | 0.4344 | 0.4242 | 0.2696 |
| | DISTS ↓ | 0.2184 | 0.2898 | 0.2067 | 0.2230 | 0.1677 | 0.2310 | 0.2156 | 0.1492 |
| | CLIP-IQA ↑ | 0.4189 | 0.4344 | 0.3494 | 0.4697 | 0.4557 | 0.2852 | 0.2417 | 0.5107 |
| | FasterVQA ↑ | 0.7386 | 0.4772 | 0.7744 | 0.6969 | 0.7489 | 0.5493 | 0.7042 | 0.8064 |
| | DOVER ↑ | 0.7060 | 0.3290 | 0.7564 | 0.7291 | 0.7264 | 0.4576 | 0.4830 | 0.7809 |
| | $E^*_{warp}$ ↓ | 4.83 | 6.12 | 3.10 | 3.97 | 3.59 | 1.03 | 2.08 | 1.77 |
| SPMCS | PSNR ↑ | 21.22 | 21.68 | 22.17 | 18.81 | 22.39 | 18.58 | 21.24 | 23.11 |
| | SSIM ↑ | 0.5613 | 0.5153 | 0.5638 | 0.4113 | 0.5896 | 0.4850 | 0.5441 | 0.6210 |
| | LPIPS ↓ | 0.3721 | 0.4467 | 0.3662 | 0.4468 | 0.3263 | 0.5358 | 0.5257 | 0.2888 |
| | DISTS ↓ | 0.2220 | 0.2697 | 0.2164 | 0.2452 | 0.1960 | 0.2669 | 0.2872 | 0.1713 |
| | CLIP-IQA ↑ | 0.5238 | 0.5442 | 0.3513 | 0.5248 | 0.4348 | 0.3188 | 0.2646 | 0.5690 |
| | FasterVQA ↑ | 0.7213 | 0.5463 | 0.7307 | 0.6556 | 0.6745 | 0.4658 | 0.4076 | 0.7245 |
| | DOVER ↑ | 0.7490 | 0.4930 | 0.6753 | 0.7171 | 0.6754 | 0.4284 | 0.3204 | 0.7828 |
| | $E^*_{warp}$ ↓ | 5.61 | 8.07 | 1.88 | 4.22 | 1.68 | 1.19 | 1.01 | 1.04 |
| YouHQ40 | PSNR ↑ | 22.82 | 23.32 | 22.39 | 19.62 | 23.17 | 19.78 | 22.64 | 24.30 |
| | SSIM ↑ | 0.6337 | 0.6273 | 0.5895 | 0.4824 | 0.6194 | 0.5911 | 0.6323 | 0.6740 |
| | LPIPS ↓ | 0.3571 | 0.4211 | 0.4091 | 0.4268 | 0.3608 | 0.4742 | 0.4600 | 0.2997 |
| | DISTS ↓ | 0.1790 | 0.2159 | 0.1933 | 0.2012 | 0.1685 | 0.2140 | 0.2287 | 0.1477 |
| | CLIP-IQA ↑ | 0.4704 | 0.4633 | 0.3964 | 0.5258 | 0.4657 | 0.3309 | 0.2739 | 0.4985 |
| | FasterVQA ↑ | 0.8401 | 0.7024 | 0.8423 | 0.8460 | 0.8363 | 0.7022 | 0.5586 | 0.8494 |
| | DOVER ↑ | 0.8572 | 0.6855 | 0.8596 | 0.8596 | 0.8446 | 0.6957 | 0.5594 | 0.8574 |
| | $E^*_{warp}$ ↓ | 5.91 | 5.75 | 3.08 | 6.84 | 3.45 | 0.95 | 2.21 | 2.05 |
| RealVSR | PSNR ↑ | 20.85 | 20.81 | 22.12 | 20.29 | 22.02 | 15.75 | 17.43 | 22.32 |
| | SSIM ↑ | 0.7105 | 0.6277 | 0.7163 | 0.5945 | 0.6774 | 0.4002 | 0.5215 | 0.7301 |
| | LPIPS ↓ | 0.2016 | 0.2312 | 0.1870 | 0.2671 | 0.2182 | 0.3784 | 0.2943 | 0.1851 |
| | DISTS ↓ | 0.1279 | 0.1435 | 0.0983 | 0.1425 | 0.1169 | 0.1688 | 0.1599 | 0.0978 |
| | CLIP-IQA ↑ | 0.7472 | 0.5553 | 0.2905 | 0.4855 | 0.4510 | 0.3880 | 0.3641 | 0.5207 |
| | FasterVQA ↑ | 0.7436 | 0.6988 | 0.7789 | 0.7403 | 0.7707 | 0.8018 | 0.7338 | 0.7959 |
| | DOVER ↑ | 0.7542 | 0.7099 | 0.7636 | 0.7114 | 0.7508 | 0.7637 | 0.7051 | 0.7867 |
| | $E^*_{warp}$ ↓ | 6.32 | 9.55 | 4.45 | 6.25 | 3.16 | 5.15 | 9.88 | 3.52 |
| MVSR4x | PSNR ↑ | 22.47 | 21.58 | 21.80 | 20.42 | 22.77 | 20.50 | 22.42 | 22.42 |
| | SSIM ↑ | 0.7412 | 0.6473 | 0.7045 | 0.6117 | 0.7418 | 0.7117 | 0.7421 | 0.7523 |
| | LPIPS ↓ | 0.4534 | 0.5945 | 0.4235 | 0.4717 | 0.3568 | 0.4471 | 0.4311 | 0.3476 |
| | DISTS ↓ | 0.3021 | 0.3351 | 0.2498 | 0.2673 | 0.2245 | 0.2800 | 0.2714 | 0.2363 |
| | CLIP-IQA ↑ | 0.4396 | 0.5003 | 0.4118 | 0.6106 | 0.3769 | 0.3104 | 0.2674 | 0.5453 |
| | FasterVQA ↑ | 0.3371 | 0.4723 | 0.7497 | 0.7663 | 0.6764 | 0.3584 | 0.2840 | 0.7742 |
| | DOVER ↑ | 0.2111 | 0.3255 | 0.6846 | 0.7221 | 0.6214 | 0.3164 | 0.2137 | 0.6984 |
| | $E^*_{warp}$ ↓ | 1.64 | 3.89 | 1.69 | 5.10 | 1.55 | 0.62 | 0.61 | 0.78 |
| VideoLQ | CLIP-IQA ↑ | 0.3617 | 0.4049 | 0.3433 | 0.4132 | 0.3465 | 0.3031 | 0.2652 | 0.3484 |
| | FasterVQA ↑ | 0.7381 | 0.5909 | 0.7586 | 0.7501 | 0.7412 | 0.6769 | 0.7028 | 0.7764 |
| | DOVER ↑ | 0.7310 | 0.6160 | 0.7388 | 0.7370 | 0.7421 | 0.6912 | 0.7080 | 0.7435 |
| | $E^*_{warp}$ ↓ | 7.58 | 7.79 | 5.97 | 13.47 | 6.79 | 6.495 | 5.96 | 5.85 |

Table 2: Quantitative comparison with state-of-the-art methods. The best and second best results are colored with red and blue. Our method outperforms on various datasets and metrics.

**Image Ratio ($\varphi$).** We further investigate the impact of the image data ratio in stage-2. The results are presented in Tab. 1b. Specifically, ratio-0% denotes training with video data only, while 100% uses only image data. Neither alone yields optimal results, since videos suffer from lower quality due to hardware limitations; while with higher quality are limited by the single-frame nature. Conversely, mixing both offers complementary strengths and improves overall performance. Based on the experiments, we finally set the image ratio to 80% (*i.e.*, $\varphi$=0.8).

**Training Dataset.** We compare different training datasets. To eliminate the influence of image data, we conduct training only in stage-1. The results are listed in Tab. 1c. For OpenVid-1M [25], we select videos with a resolution higher than 1080p (1080×1920), resulting in approximately 0.4M videos. The YouHQ dataset [63] contains 38,576 1080p videos. We observe substantial performance variation across datasets. Notably, our proposed HQ-VSR dataset achieves superior performance despite containing 2,055 videos, which is significantly fewer than the other datasets.

**Video Processing Pipeline.** We also performed an ablation on the proposed video processing pipeline. The results are presented in Tab. 1d. First, we apply the filtering steps (*i.e.*, +Filter), including metadata, scene, and quality filtering) to the raw dataset (*i.e.*, OpenVid-1M [25]). Although OpenVid has undergone some preprocessing, applying more filtering tailored to VSR further improves data quality. Then, we apply motion processing on the filtered data to exclude static videos and regions. This leads to further performance gains, confirming the benefit of motion detection cropping.

### 4.3 Comparison with State-of-the-Art Methods

We compare our efficient one-step diffusion model, DOVE, with recent state-of-the-art image and video super-resolution methods: RealESRGAN [38], ResShift [56], RealBasicVSR [5], Upscale-A-Video [63], MGLD-VSR [50], VEnhancer [9], and STAR [48].

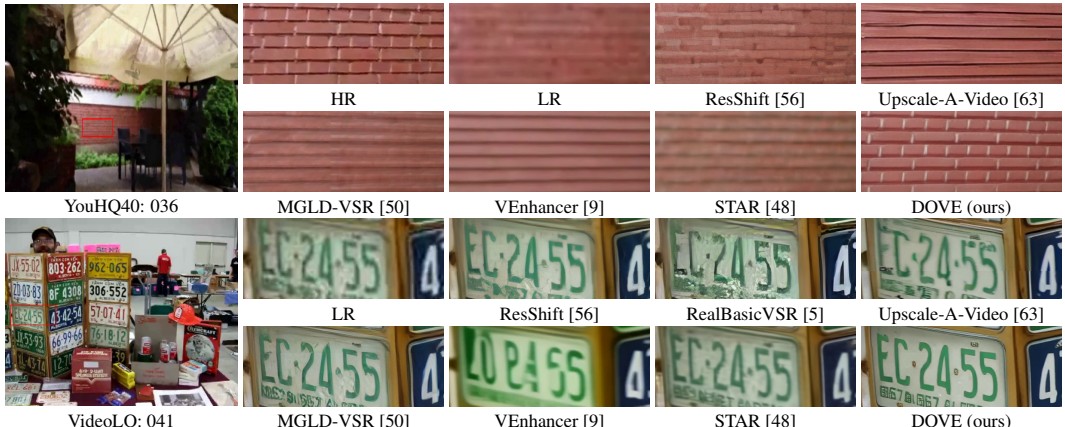

Figure 4: Visual comparison on synthetic (YouHQ40 [63]) and real-world (VideoLQ [5]) datasets. The videos in VideoLQ are sourced from the Internet without high-resolution (HQ) references.

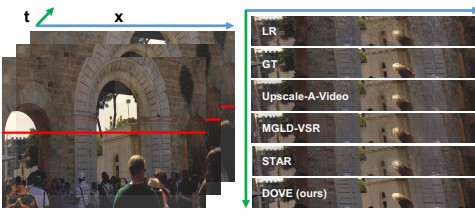

Figure 5: Comparison of temporal consistency (stacking the red line across frames).

| Method | Step | Time (s) | Performance |
|---|---|---|---|
| Upscale-A-Video [63] | 30 | 279.32 | 0.7370 |
| MGLD-VSR [50] | 50 | 425.23 | 0.7421 |
| VEnhancer [9] | 15 | 121.27 | 0.6912 |
| STAR [48] | 15 | 173.07 | 0.7080 |
| **DOVE (ours)** | **1** | **14.90** | **0.7435** |

Table 3: Comparison of inference step (Step), running time (Time), and DOVER on VideoLQ (Performance) of different diffusion-based methods.

**Quantitative Results.** We present quantitative comparisons in Tab. 2. Our DOVE achieves outstanding performance across diverse datasets. For fidelity metrics (*i.e.*, PSNR and SSIM), our model achieves the best performance on most datasets. For perceptual metrics (*i.e.*, LPIPS, DISTS, and CLIP-IQA), the DOVE ranks first or second on five datasets. Furthermore, for video-specific metrics regarding quality (*i.e.*, FasterVQA and DOVER) and consistency (*e.g.*, $E_{warp}^*$), the DOVE also performs strongly. **Besides, we provide more comparison results in the supplementary material.**

**Qualitative Results.** We provide visual comparisons on both synthetic (*i.e.*, YouHQ40) and real-world (*i.e.*, VideoLQ) videos in Fig. 4. Our DOVE produces more realistic results. For instance, in the first case, DOVE successfully reconstructs the brick pattern, while other methods yield overly smooth or inaccurate results. Similarly, in the second example, our method delivers sharper restoration. More visual results are provided in the supplementary material.

**Temporal Consistency.** We also visualize the temporal profile in Fig. 5. We can observe that existing methods struggle under complex degradations, exhibiting misalignment (*e.g.*, Upscale-A-Video [63] and STAR [48]) or blurring (*e.g.*, MGLD-VSR [50]). In contrast, our method achieves excellent temporal consistency, with smooth transitions and rich details across frames. This is due to the strong prior of the pretrained model and our effective latent-space training strategy.

**Running Time Comparisons.** We compare the inference step (Step), running time (Time), and performance (DOVER on VideoLQ [5]) of different diffusion-based video super-resolution methods in Tab. 3. For fairness, all methods are measured running time on the same A100 GPU, generating a 33-frame 720×1280 video. Our method is approximately **28×** faster than MGLD-VSR [50]. Even compared with the fastest compared method, VEnhancer [9], the DOVE is **8** times faster. More comprehensive analyses are provided in the supplementary material.

## 5 Conclusion

In this paper, we propose an efficient one-step diffusion model, DOVE, for real-world video super-resolution (VSR). DOVE is constructed based on the pretrained video generation model, CogVideoX. To enable effective fine-tuning, we introduce the latent-pixel training strategy. It is a two-stage scheme that gradually adapts the pretrained video model to the VSR task. Moreover, we construct a high-quality dataset, HQ-VSR, to further enhance performance. The dataset is generated by our proposed video processing pipeline, which is tailored for VSR. Extensive experiments demonstrate that our DOVE outperforms state-of-the-art methods with high efficiency.

## Acknowledgments

This work was supported by Shanghai Municipal Science and Technology Major Project (2021SHZDZX0102), the Fundamental Research Funds for the Central Universities, the Special Project on Technological Innovation Application for the 15th National Games, and the National Paralympic Games under Grant 2025B01W0005.

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
