# OpenReview forum: "DOVE: Efficient One-Step Diffusion Model for Real-World Video Super-Resolution"
_NeurIPS.cc/2025/Conference — NeurIPS 2025 poster_

### Official Review · Reviewer_svnJ · 2025-06-27

**Clarity:** 3
**Significance:** 3
**Originality:** 2
**Rating:** 4
**Confidence:** 5

**Summary:**

The paper proposes an efficient one-step diffusion model, DOVE, based on CogVideoX for real-world video super-resolution. The DOVE introduces no extra modules but keeps the simple CogVideoX's Encoder-DiT-Decoder schema. To adapt the T2V-based CogVideoX to SR task, the DOVE employs a two-stage training with high-quality video obtained by a video processing pipeline.

**Questions:**

- My main question is about the utilization of diffusion priors. Since the DOVE is simply adapted from commonly used perceptual VSR/ISR losses and does not use CogVideoX to calibrate the output, I have doubts about identifying the DOVE as a "diffusion" model. For instance, replacing the CogVedeo with a GAN,  like GigaGAN to VideoGigaGAN, the pipeline still works. It seems like the improvement is brought by the scaling laws of changing the base model from weak GigaGAN to powerful CogVideoX.
- In Table 1 (b), the DOVE (80%) leads by DOVE (100%) in a rather small margin. However, Table 2 compares only ResShift, which is a rather weak baseline even for image SR. Comparison with more recent diffusion-based SR and GAN-based VSR will raise my confidence in DOVE.
- The visual results in sup tend to be extremely sharp and lose details, making it look unnatural. Moreover, the high sharpness and smoothness will induce higher subjective scores.

**Ethical Concerns:**

["NO or VERY MINOR ethics concerns only"]

**Final Justification:**

The rebuttal addressed most of my concerns.

**Limitations:**

yes

**Quality:**

3

**Strengths And Weaknesses:**

Strength:
- The overall pipeline is simple but effective. The DOVE, by avoiding introducing complicated modules, can achieve higher efficiency.
- The video processing pipeline is interesting and ought to work for most video restoration tasks.
- The experimental results are satisfactory to show DOVE advancing recent diffusion-based VSR.

Weakness:
- Despite the simplicity and intuitiveness, the originality is limited. The DOVE is simply adapted from commonly used perceptual VSR/ISR losses. It barely uses the abundant diffusion priors from CogVideoX except for initialization.
- The visual results tend oversharp and smooth.

---

> ### Author Rebuttal · Authors · 2025-07-29
>
> # Response to Reviewer svnJ (denoted as R4)
>
> `Q4-1` Despite the simplicity and intuitiveness, the originality is limited. The DOVE is simply adapted from commonly used perceptual VSR/ISR losses. It barely uses the abundant diffusion priors from CogVideoX except for initialization.
>
> `A4-1` Thank you for the comment. We clarify our **novelty** and **prior** utilization below.
>
> **(1) Novelty Clarification.**
>
> 1. **Common Loss.** The commonly used loss functions, such as MSE, DISTS, and Frame losses, are not our novelty. We adopt these commonly used losses to ensure **general applicability**.
>
> 2. **(Novelty) Training Strategy.** Our main contribution lies in the proposed **latent-pixel training strategy**, which consists of two stages. Stage-1 learns the one-step mapping in the latent space. Stage-2 enhances reconstruction in pixel-space. Additionally, we introduce **image-video mixed training** and a **frame-by-frame VAE encoder-decoder** to better adapt to VSR training.
>
> 3. **Experiment Results.** Table 1a in the main paper demonstrates the effectiveness of our training strategy. We provide the results below:
>
>    | Training Stage |  Stage-1  | Stage-1+Stage-2(Image) | Stage-1+Stage-2(Image/Video) |
>    | -------------- | :-------: | :--------------------: | :--------------------------: |
>    | PSNR↑          | **27.20** |         26.39          |            26.48             |
>    | LPIPS↓         |  0.3037   |         0.2784         |          **0.2696**          |
>    | CLIP-IQA↑      |  0.3236   |         0.5085         |          **0.5107**          |
>    | DOVER↑         |  0.6154   |         0.7694         |          **0.7809**          |
>
> 4. **(Another Novelty) Data Pipeline.** We also design a **video processing pipeline** to construct the HQ-VSR dataset. This dataset also contributes to the performance.
>
>
>
> **(2) Diffusion Prior.**
>
> 1. **Prior Utilization.** We apply pretrained diffusion priors for initialization to enhance the restoration capability of DOVE.
> 2. **Task Gap.** Since the original model (*i.e.*, CogVideoX) is designed for generation rather than restoration, we propose the **latent-pixel training strategy** to better adapt it to the VSR task.
> 3. **Effective Training Design.** Our design ensures architectural simplicity, training efficiency, and impressive performance.
>
>
>
> `Q4-2` The visual results tend oversharp and smooth.
>
> `A4-2` Thank you for the comment. In some challenging cases, the visual results of our DOVE may appear oversharpened and smooth. However, this typically occurs in highly degraded scenarios where existing methods struggle to recover the right structure.
>
> We provide several examples in the **supplementary material** to illustrate this point.
>
> - **Case 1** *(Fig. 5, first sample, UDM10: 000)*: The LR input is heavily blurred, and the wall textures are barely visible. Our DOVE successfully reconstructs the correct structural lines, although slightly oversharpened. In comparison, other methods fail to recover the lines and even introduce artifacts.
> - **Case 2** *(Fig. 7, fourth sample, VideoLQ: 016)*: While DOVE produces a slightly oversharpened result, it restores the text correctly and without artifacts. In contrast, methods like STAR misrecognize the word "CRIPPLE" as "CHIPPLE", while other methods produce artifacts.
>
> **Overall**, DOVE achieves superior restoration quality compared to existing methods. Of course, we acknowledge the reviewer's observation that some challenging cases may exhibit oversharpening or smoothing. This is an important direction for future improvement.
>
>
>
> `Q4-3` My main question is about the utilization of diffusion priors. Since the DOVE is simply adapted from commonly used perceptual VSR/ISR losses and does not use CogVideoX to calibrate the output, I have doubts about identifying the DOVE as a "diffusion" model. For instance, replacing the CogVedeo with a GAN, like GigaGAN to VideoGigaGAN, the pipeline still works. It seems like the improvement is brought by the scaling laws of changing the base model from weak GigaGAN to powerful CogVideoX.
>
> `A4-3` Thank you for the question. Our approach is a diffusion model. We explain it below.
>
> **(1) Following the Diffusion Paradigm.**
>
> 1. **Diffusion Formulation.** DOVE follows the core principle of diffusion models: generating results through denoising. Specifically, our model adopts the v-prediction scheme, where a Transformer predicts the velocity (a variant of noise). The restored latent **${z}_{sr}$** is obtained using:
>
>    $z_{sr} = \sqrt{ᾱ_t} z_{lr} - \sqrt{1-ᾱ_t} v_θ(z_{lr}, t)$.
>
>    This formulation is consistent with the definition of the diffusion model.
>
> 2. **Consistency with Existing Methods.** Our method aligns with the framework of other one-step diffusion models (*e.g.*, those discussed by **Reviewer-WLko** in `Q3-1`: UltraVSR [1], DLoRAL [2], and OSEDiff [3]). From the framework definition perspective, the main difference between these diffusion methods and our DOVE is the loss function, but this does not affect the overall definition.
>
>
>
> **(2) Pipeline Transferability.**
>
> 1. The pipeline is transferable. One strength of our method is its **generalizability**. The proposed training strategy can be adapted to other pretrained T2V backbones, not just CogVideoX.
>
> 2. Although pipeline transfer is possible, **diffusion priors** are central to our approach. Compared to GANs, diffusion models offer more stable training and stronger generative priors. These properties contribute significantly to the impressive performance of DOVE.
>
> 3. Motivated by the reviewer's comment, we extend DOVE to another pretrained diffusion T2V model, **WAN2.1-T2V-1.3B** [**ref1**]. We refer to the resulting model as **DOVE (WAN2.1)**. The evaluation results on UDM10 are as follows:
>
>    | Method          |   PSNR↑   |   LPIPS↓   | CLIP-IQA↑  |   DOVER↑   |
>    | --------------- | :-------: | :--------: | :--------: | :--------: |
>    | Upscale-A-Video |   21.72   |   0.4116   |   0.4697   |   0.7291   |
>    | MGLD-VSR        |   24.23   |   0.3272   |   0.4557   |   0.7264   |
>    | STAR            |   23.47   |   0.4242   |   0.2417   |   0.4830   |
>    | DOVE (WAN2.1)   | **25.28** | **0.2555** | **0.5133** | **0.7563** |
>
>    The WAN-based model shows strong VSR performance, validating the generality and effectiveness of our method.
>
>
>
> [**ref1**] Wan: Open and Advanced Large-Scale Video Generative Models, 2025.
>
>
>
> `Q4-4` In Table 1 (b), the DOVE (80%) leads by DOVE (100%) in a rather small margin. However, Table 2 compares only ResShift, which is a rather weak baseline even for image SR. Comparison with more recent diffusion-based SR and GAN-based VSR will raise my confidence in DOVE.
>
> `A4-4` Thank you for the feedback. We provide detailed clarifications below.
>
> **(1) Improvement Clarification.** Compared with DOVE (100%), DOVE (80%) has a quantitative improvement of 0.0115 in the DOVER metric, which is not very small. Meanwhile, image-video mixed training leads to more stable and consistent results.
>
>
>
> **(2) More Comparison Methods.** We include comparisons with more recent methods:
>
> - **SinSR** [**ref2**]: Image SR, diffusion-based, CVPR 2024
> - **OSEDiff** [**ref3**]: Image SR, diffusion-based, NeurIPS 2024
> - **RealViformer** [**ref4**]: Video SR, GAN-based, ECCV 2024
>
> (Note: VideoGigaGAN is not publicly released, so we cannot include it in the evaluation.)
>
> The comparison results are as follows:
>
> | Dataset                     |   UDM10    |                   |   SPMCS    |                   |  RealVSR   |                   |   MVSR4x   |                   |
> | --------------------------- | :--------: | :---------------: | :--------: | :---------------: | :--------: | :---------------: | :--------: | :---------------: |
> | **Method**                  | **DOVER↑** | **$E^*_{warp}$↓** | **DOVER↑** | **$E^*_{warp}$↓** | **DOVER↑** | **$E^*_{warp}$↓** | **DOVER↑** | **$E^*_{warp}$↓** |
> | SinSR (SR, CVPR'24)         |   0.4091   |       8.59        |   0.6052   |       11.20       |   0.7192   |       11.97       |   0.3974   |       5.86        |
> | OSEDiff (SR, NeurIPS'24)    |   0.7240   |       5.73        |   0.7380   |       4.77        |   0.7650   |       7.21        |   0.6729   |       2.75        |
> | RealViformer (VSR, ECCV'24) |   0.7303   |       2.29        |   0.5905   |       1.46        |   0.7191   |       3.93        |   0.6451   |       1.25        |
> | DOVE (ours)                 | **0.7809** |     **1.77**      | **0.7828** |     **1.04**      | **0.7867** |     **3.52**      | **0.6984** |     **0.78**      |
>
> The results show that DOVE outperforms recent diffusion- and GAN-based methods in video quality and temporal consistency.
>
>
>
> [**ref2**] SinSR: Diffusion-Based Image Super-Resolution in a Single Step, CVPR 2024.
>
> [**ref3**] One-Step Effective Diffusion Network for Real-World Image Super-Resolution, NeurIPS 2024.
>
> [**ref4**] RealViformer: Investigating Attention for Real-World Video Super-Resolution, ECCV 2024.
>
>
>
> `Q4-5` The visual results in sup tend to be extremely sharp and lose details, making it look unnatural. Moreover, the high sharpness and smoothness will induce higher subjective scores.
>
> `A4-5` Thank you for the comment.
>
> In challenging cases with severe degradation, DOVE may produce results that appear oversharpened or overly smooth. However, it still outperforms existing methods regarding fidelity and visual quality.
>
> Additionally, we agree with the reviewer that better balancing sharpness and detail preservation is a valuable direction for future work.
>
> For more details, please see `A4-2`.

---

> > ### Comment · Reviewer_svnJ · 2025-08-04
> > **Re**
> >
> > The rebuttal addressed most of my concerns, but I still have a question on diffusion prior. Depite using $z_{sr} = \sqrt{ᾱ_t} z_{lr} - \sqrt{1-ᾱ_t} v_θ(z_{lr}, t)$ to obtain the results, DOVE dosen't show implicit use of diffusion priors liker VSD or directly use diffusion loss to make sure the output in the domain of diffusion, making the training process more like a GAN. Moreover, compared to LoRA-based methods, DOVE fully trains DiT, which may further lead to loss of pre-training information.

---

> > > ### Author Response · Authors · 2025-08-04
> > > **Further Discussion on Diffusion Prior**
> > >
> > > Dear Reviewer svnJ,
> > >
> > > Thank you for your response.
> > >
> > > First, we are glad that our rebuttal has addressed most of your concerns.
> > >
> > > Second, regarding the usage of diffusion priors, we would like to provide a more detailed explanation:
> > >
> > > 1. We use the pretrained diffusion model for initialization, which provides strong prior knowledge. It is reflected in the strong performance of our method.
> > > 2. The pretrained model (e.g., CogVideoX) is designed for video generation, while our DOVE is tailored for VSR. Considering the task difference, we avoid constraining the output space. Our design allows DOVE to surpass the performance limitations of the original generative model.
> > > 3. We think the definition of the one-step diffusion model should remain broad and not be limited to specific loss types such as VSD. Some previous one-step diffusion works (e.g., [**ref1**, **ref2**]) also do not rely on implicit objectives like VSD. In our view, any method that follows the diffusion model formulation and utilizes diffusion priors qualifies as a  (generalized) diffusion model.
> > > 4. The VSD loss in some methods (e.g., OSEDiff) is primarily used as a performance booster, similar to LPIPS or adversarial (GAN) losses. Its use is optional and does not define whether a method is diffusion-based.
> > > 5. We appreciate reviewers' suggestion to use more diffusion prior. In future work, we will explore more efficient and principled ways of leveraging diffusion priors. (PS: The overhead of VSD is substantial, making it less suitable for efficient VSR training.)
> > >
> > > Finally, we sincerely thank you again for your thoughtful comments. We truly appreciate your insights and look forward to your response.
> > >
> > >
> > >
> > >
> > >
> > > [**ref1**] SF-V: Single Forward Video Generation Model, NeurIPS 2024.
> > >
> > > [**ref2**] OSDFace: One-Step Diffusion Model for Face Restoration, CVPR 2025.
> > >
> > >
> > >
> > > Best regards,
> > >
> > > Authors

---

> > > > ### Comment · Reviewer_svnJ · 2025-08-04
> > > > **Re**
> > > >
> > > > I appreciate your quick response. I have no more questions.

---

> > > > > ### Author Response · Authors · 2025-08-04
> > > > > **Thanks Reviewer svnJ for approving our work**
> > > > >
> > > > > Dear Reviewer svnJ,
> > > > >
> > > > > Thank you for your timely response and for acknowledging our work. We sincerely appreciate your valuable suggestions and will refine our paper based on your feedback.
> > > > >
> > > > > Best regards,
> > > > >
> > > > > Authors

---

### Official Review · Reviewer_WLko · 2025-07-02

**Clarity:** 4
**Significance:** 3
**Originality:** 3
**Rating:** 5
**Confidence:** 4

**Summary:**

An efficient one-step diffusion model DOVE was proposed to achieve the real-world video super resolution. A latent–pixel training strategy was designed to effectively train DOVE and a video processing pipeline was proposed to construct a high-quality dataset to enhance the restoration capability. Extensive experiments show that DOVE exhibits comparable or superior performance.

**Questions:**

Comparison of Fairness

DOVE model proposed the video processing pipeline to fine-tuning the VSR models on high-quality datasets. In state-of-the-art VSR comparison in Table 2, the DOVE model using the proposed video processing pipeline achieves the best performance in most scenarios, while other methods without used.  Can other existing methods also use this pipeline to improve performance? The more results and discussion should be provided.

**Ethical Concerns:**

["NO or VERY MINOR ethics concerns only"]

**Final Justification:**

After reading the author's rebuttal, the detailed explanation and experimental results are provided, my concerns were addressed. I decide my score as accept.

**Limitations:**

Please see weaknesses and questions.

**Quality:**

4

**Strengths And Weaknesses:**

Strengths

The first one-step diffusion model was proposed for VSR task, this model is at least 11× faster over existing diffusion-based VSR models. The efficient latent-pixel training strategy and a video processing pipeline were developed to enable effective fine-tuning proposed model. These designs make the DOVE achieve the state-of-the-art performance on multiple benchmarks.


Weaknesses

Insufficient novelty.
One-step diffusion models were proposed in many existing methods, i.e., [1]-[4]. The difference between the proposed DOVE and existing methods should be clarified and discussed. Additionally, in video processing pipeline, the metadata/scene/quality filtering were proposed by existing methods, i.e., Stable video diffusion, Openvid-1m, Open-sora. The DOVE only added more metrics, which is insufficient novelty.

[1] Liu, Y., Pan, J., Li, Y., Dong, Q., Zhu, C., Guo, Y. and Wang, F., 2025. UltraVSR: Achieving Ultra-Realistic Video Super-Resolution with Efficient One-Step Diffusion Space. arXiv preprint arXiv:2505.19958.

[2] Sun, Y., Sun, L., Liu, S., Wu, R., Zhang, Z. and Zhang, L., 2025. One-Step Diffusion for Detail-Rich and Temporally Consistent Video Super-Resolution. arXiv preprint arXiv:2506.15591.

[3] Wu, R., Sun, L., Ma, Z. and Zhang, L., 2024. One-step effective diffusion network for real-world image super-resolution. Advances in Neural Information Processing Systems, 37, pp.92529-92553.

[4] Wang, Y., Yang, W., Chen, X., Wang, Y., Guo, L., Chau, L.P., Liu, Z., Qiao, Y., Kot, A.C. and Wen, B., 2024. Sinsr: diffusion-based image super-resolution in a single step. In Proceedings of the IEEE/CVF conference on computer vision and pattern recognition (pp. 25796-25805).


Experimental results.
In Table 1-3 provide the corresponding results that proves the effectiveness of DOVE model. Some detailed results should be provided. For example, The image ratio interval in Table 1 (b) is too large, and setting a smaller interval is beneficial for obtaining the optimal ratio. The experimental results of each filter operation in video processing pipeline should be provided in Table 1(d) for verifying the effectiveness of each filter.  Besides, in Table 1 (d), the "+filter" (steps 2-4) only brings a PSNR gain of 0.05dB. How to verify the effectiveness of the proposed filters?

---

> ### Author Rebuttal · Authors · 2025-07-29
>
> # Response to Reviewer WLko (denoted as R3)
>
> `Q3-1` Insufficient novelty. One-step diffusion models were proposed in many existing methods, i.e., [1]-[4]. The difference between the proposed DOVE and existing methods should be clarified and discussed. Additionally, in video processing pipeline, the metadata/scene/quality filtering were proposed by existing methods, i.e., Stable video diffusion, Openvid-1m, Open-sora. The DOVE only added more metrics, which is insufficient novelty.
>
> `A3-1` Thank you for the comment. We clarify the novelty of our **one-step** model and data **pipeline** below.
>
> **(1) One-Step Novelty.**
>
> ***General Difference.***
>
> 1. OSEDiff [3] and SinSR [4] are designed for **image** SR and are not tailored to the video task. Meanwhile, UltraVSR [1] and DLoRAL [2] are one-step VSR methods released on arXiv **after** the NeurIPS submission deadline.
> 2. Existing methods [1]–[4] rely on pretrained **image** models and incorporate specifically designed modules to enhance performance.
> 3. In contrast, our method is built upon a pretrained **video** model. We propose a well-designed **training strategy** that activates and enhances the VSR ability of the pretrained model.
>
> ***Detailed Difference.***
>
> 1. **UltraVSR [1]** enhances temporal consistency by inserting temporal layers into a pretrained **image** model (*i.e.*, stable diffusion) and uses a degradation-aware **module** to improve performance.
> 2. **DLoRAL [2]** also relies on the **image** model (stable diffusion) uses a dual-stage process to train consistency-LoRA and detail-LoRA modules. However, it only supports two-frame inference.
> 3. **OSEDiff [3]** is an **image** SR method using LoRA and variational score distillation, leveraging pretrained image priors.
> 4. **SinSR [4]** distills a multi-step **image** SR model (*i.e.*, ResShift) into a one-step student within the image domain.
> 5. **DOVE (ours)** utilizes a pretrained **video** model and introduces the **latent-pixel training strategy** designed explicitly for VSR, achieving high-quality one-step restoration.
>
> ***Comparison Results.***
>
> We compare DOVE against existing one-step SR methods: OSEDiff, SinSR, and DLoRAL (UltraVSR code is not publicly available). The comparison results are as follows:
>
> | Dataset     |   UDM10    |                   |   SPMCS    |                   |  RealVSR   |                   |   MVSR4x   |                   |
> | ----------- | :--------: | :---------------: | :--------: | :---------------: | :--------: | :---------------: | :--------: | :---------------: |
> | **Method**  | **DOVER↑** | **$E^*_{warp}$↓** | **DOVER↑** | **$E^*_{warp}$↓** | **DOVER↑** | **$E^*_{warp}$↓** | **DOVER↑** | **$E^*_{warp}$↓** |
> | SinSR [4]   |   0.4091   |       8.59        |   0.6052   |       11.20       |   0.7192   |       11.97       |   0.3974   |       5.86        |
> | OSEDiff [3] |   0.7240   |       5.73        |   0.7380   |       4.77        |   0.7650   |       7.21        |   0.6729   |       2.75        |
> | DLoRAL [2]  |   0.7219   |       3.40        |   0.6741   |       2.98        |   0.7539   |       4.73        |   0.6267   |       1.60        |
> | DOVE (ours) | **0.7809** |     **1.77**      | **0.7828** |     **1.04**      | **0.7867** |     **3.52**      | **0.6984** |     **0.78**      |
>
> We observe that **image SR methods** (OSEDiff and SinSR) perform poorly on video-specific metrics, *e.g.*, DOVER or $E^*_{warp}$, confirming their unsuitability for VSR tasks. Compared to DLoRAL, our DOVE achieves superior performance, validating the effectiveness of our method.
>
>
>
> **(2) Pipeline Novelty.**
>
> 1. **Metadata Filtering Difference.** Unlike previous pipelines, our process emphasizes high-resolution video, which is essential for VSR and not explicitly focused on in other pipelines.
> 2. **Quality Filtering Difference.** As the eviewer (WLko) noted, we incorporate more diverse quality metrics. This is consistent with the goal of VSR, which is to recover fine-grained details.
> 3. **Motion Filtering Difference.** This step is unique to our pipeline. This fits the characteristics of VSR clipping training and is not included in previous pipelines.
> 4. **Ablation Results.** Table 1d in the main paper demonstrates the effectiveness of the filtering and motion process. Further ablation for each filter component is provided in `A3-2`.
> 5. **Comparison with Openvid-1M.** As shown in Tab. 1c (main paper), models trained on HQ-VSR outperform those trained on the existing dataset, Openvid-1M. This demonstrates the value of our video processing pipeline.
>
>
>
> `Q3-2` Experimental results. In Table 1-3 provide the corresponding results that proves the effectiveness of DOVE model. Some detailed results should be provided. For example, The image ratio interval in Table 1 (b) is too large, and setting a smaller interval is beneficial for obtaining the optimal ratio. The experimental results of each filter operation in video processing pipeline should be provided in Table 1(d) for verifying the effectiveness of each filter. Besides, in Table 1 (d), the "+filter" (steps 2-4) only brings a PSNR gain of 0.05dB. How to verify the effectiveness of the proposed filters?
>
> `A3-2` Thank you for the suggestions. We conduct additional ablations for the image ratio and each filtering step. We also clarify the filter gains.
>
>
>
> **(1) Ablation: Fine-Grained Image Ratio.**
>
> We conduct more fine-grained experiments on the image ratio. Based on the preliminary results in Tab. 1b, we compare the cases of 70%, 75%, 80%, 85%, and 90%, and the results are as follows:
>
> | Image Ratio |  70%   |  75%   |    80%     |    85%     |  90%   |
> | ----------- | :----: | :----: | :--------: | :--------: | :----: |
> | PSNR↑       | 26.33  | 26.36  |   26.48    | **26.54**  | 26.39  |
> | LPIPS↓      | 0.2713 | 0.2680 |   0.2696   | **0.2644** | 0.2715 |
> | CLIP-IQA↑   | 0.5053 | 0.5074 | **0.5107** |   0.5083   | 0.5023 |
> | DOVER↑      | 0.7785 | 0.7801 |   0.7809   | **0.7870** | 0.7776 |
>
> We find that 80% and 85% yield a good balance between frame fidelity and perceptual quality. This is close to the settings (80%) in our paper. For consistency, we still adopt the image ratio of 80%.
>
>
>
> **(2) Ablation: Data Filtering Steps.**
>
> We perform a **breakdown** ablation of the three filtering components in our data pipeline: Metadata, Scene, and Quality filtering. The results are as follows:
>
> | Method            |   PSNR↑   |   LPIPS↓   | CLIP-IQA↑  |   DOVER↑   |
> | ----------------- | :-------: | :--------: | :--------: | :--------: |
> | OpenVid-1M        |   27.04   |   0.3376   |   0.2683   |   0.4363   |
> | + Metadata Filter | **27.22** |   0.3368   |   0.2765   |   0.4511   |
> | + Scene Filter    |   27.20   |   0.3363   |   0.2771   |   0.4529   |
> | + Quality Filter  |   27.09   | **0.3236** | **0.2894** | **0.5357** |
>
> Different filters have different functions:
>
> - **Metadata filtering** removes low-resolution videos that are unsuitable for VSR. This improves fidelity (*e.g.*, PSNR) and slightly enhances perceptual quality.
> - **Scene filtering** has a limited effect since OpenVid-1M already includes scene cuts. However, it is essential to extend the pipeline to other unprocessed data.
> - **Quality filtering** significantly enhances perceptual metrics, particularly DOVE. The slight drop in PSNR may result from imperfect alignment between restored details and the ground truth. But it still performs better than the unfiltered dataset (*i.e.*, OpenVid-1M).
>
>
>
> **(3) Clarification on Filter Effectiveness.**
>
> 1. **Caption Correction:** The label, "+Filter (steps 2–4)"  in Tab. 1 (main paper) is a typo. Actually, it is "+Filter (steps 1-3)". We will revise it.
> 2. **Filter-Specific Effects:** As shown above, each filter contributes differently to fidelity and perceptual performance.
> 3. **Perceptual Improvement:** The primary purpose of filtering is to improve perceptual quality. While the overall PSNR gain is 0.05 dB, perceptual improvements are significant. For example, DOVER improves by 0.0994.
>
>
>
> `Q3-3` Comparison of Fairness: DOVE model proposed the video processing pipeline to fine-tuning the VSR models on high-quality datasets. In state-of-the-art VSR comparison in Table 2, the DOVE model using the proposed video processing pipeline achieves the best performance in most scenarios, while other methods without used. Can other existing methods also use this pipeline to improve performance? The more results and discussion should be provided.
>
> `A3-3` Thank you for the comment.
>
> 1. Our training strategy is one of the innovations in our method. Therefore, we directly compare DOVE with existing approaches that do not use our training strategy.
>
> 2. The training strategy of existing diffusion VSR methods is tightly bound to model design. Thus, directly applying our training strategy to those models is not easy and requires extensive adaptation.
>
> 3. But our method is well-suited for pretrained text-to-video (T2V) models. To reveal the generalizability of our method, we conduct experiments using another pretrained T2V model, **WAN2.1-T2V-1.3B** [**ref1**], with the same training settings as DOVE. We refer to the resulting model as **DOVE (WAN2.1)**.
>
>    The evaluation results on UDM10 are listed below:
>
>    | Method          |   PSNR↑   |   LPIPS↓   | CLIP-IQA↑  |   DOVER↑   |
>    | --------------- | :-------: | :--------: | :--------: | :--------: |
>    | Upscale-A-Video |   21.72   |   0.4116   |   0.4697   |   0.7291   |
>    | MGLD-VSR        |   24.23   |   0.3272   |   0.4557   |   0.7264   |
>    | STAR            |   23.47   |   0.4242   |   0.2417   |   0.4830   |
>    | DOVE (WAN2.1)   | **25.28** | **0.2555** | **0.5133** | **0.7563** |
>
>    The WAN-based model also achieves strong VSR performance, further confirming the generalizability of our proposed method.
>
>
>
> [**ref1**] Wan: Open and Advanced Large-Scale Video Generative Models, 2025.

---

> > ### Comment · Reviewer_WLko · 2025-08-06
> > **Review after rebuttal**
> >
> > Thanks authors provided the detailed explanation and experimental results. My concerns have been addressed, and I decide to raise my score.

---

> > > ### Author Response · Authors · 2025-08-06
> > > **Thanks Reviewer WLko for approving our work**
> > >
> > > Dear Reviewer WLko,
> > >
> > > Thank you for your response. We are glad that our rebuttal has addressed your concerns, and we truly appreciate your time and valuable feedback. We will incorporate your suggestions to further improve the paper in the revision.
> > >
> > > Best regards,
> > >
> > > Authors

---

> ### Author Response · Authors · 2025-08-06
> **Further discussion with Reviewer WLko**
>
> Dear Review WLko,
>
> We sincerely appreciate the time and effort you devoted to reviewing our work, as well as your thoughtful and insightful comments. We have carefully considered your feedback and provided detailed responses to your concerns:
>
> 1. We clarify the novelty of our training strategy and dataset pipeline.
> 2. We conduct more fine-grained ablations on the image ratio and data filtering components.
> 3. We extend our method to other backbones to demonstrate its generalizability.
>
> As the author–reviewer discussion phase is nearing its conclusion, we would like to confirm whether our responses have addressed your concerns. If you have any questions or suggestions, please do not hesitate to let us know.
>
> Thank you again for your thoughtful feedback and kind support.
>
> Best regards,
>
> Authors

---

### Official Review · Reviewer_Xx99 · 2025-07-02

**Clarity:** 3
**Significance:** 2
**Originality:** 1
**Rating:** 4
**Confidence:** 3

**Summary:**

The paper presents DOVE, a one-step diffusion model designed for real-world video super-resolution. DOVE builds on the pre-trained diffusion model CogVideoX and introduces two key contributions. First, a two-stage training strategy is proposed: the model is initially adapted using latent representations obtained by the frozen encoders (and the diffusion encoder), followed by training on a combination of images and video frames. Second, the authors introduce a new Video High-Resolution dataset (HQ-VSR), which improves the performance of the downstream task.

**Questions:**

* Table 1c provides an initial indication of the significance of the proposed HQ-VSR dataset. My question focuses on two key aspects: how much the performance of the proposed model degrades without training or adaptation using HQ-VSR, and to what extent related work or baseline methods might improve if they were evaluated using the same dataset.

* L_S2 has three components (MSE, Dists, and frame).  How does the model perform if one or two elements are removed?

**Ethical Concerns:**

["NO or VERY MINOR ethics concerns only"]

**Final Justification:**

After reading the author's rebuttal, my concerns were fully addressed. I upvoted to weak accept

**Limitations:**

yes

**Paper Formatting Concerns:**

there is no problem with the paper formatting

**Quality:**

2

**Strengths And Weaknesses:**

The paper has the following strengths based on the model description and results:
* The model effectively multiple representations of low-resolution video, including latent space features of a pretrained diffusion model as a first step to the downstream task. Then, the model combines both image reconstructions as a single unit and frame differences as sequence information.

* The results show outperforming several baselines in both quantitative metrics and qualitative results, considering that the model is a one-step diffusion.

The only weakness is related to novelty.  L_S1 and L_S2 have elements that have already been applied. In the case of L_S1, comparing the latent representations between the prediction and the ground truth is not totally new (i.e., StableSR or DiffBIR).  For the case of L_S2, this stage incorporates established components like pixel-space MSE loss and perceptual loss. [1] uses the differences between frames.

[1] Isobe et al. (2022). Look Back and Forth: Video Super‑Resolution with Explicit Temporal Difference Modeling (CVPR 2022)

---

> ### Author Rebuttal · Authors · 2025-07-29
>
> # Response to Reviewer Xx99 (denoted as R2)
>
> `Q2-1` The only weakness is related to novelty. L_S1 and L_S2 have elements that have already been applied. In the case of L_S1, comparing the latent representations between the prediction and the ground truth is not totally new (i.e., StableSR or DiffBIR). For the case of L_S2, this stage incorporates established components like pixel-space MSE loss and perceptual loss. [1] uses the differences between frames.
>
> `A2-1` Thank you for the comment. We clarify our **novelty** as follows:
>
> 1. **Loss Function.** The specific loss functions, such as MSE, DISTS, and Frame losses, are not claimed to be innovative. We adopt these standard losses to ensure general applicability.
>
> 2. **(Novelty) Training Strategy.** Our core contribution is the proposed **latent-pixel training strategy**. Unlike common latent-space losses (*e.g.*, StableSR, DiffBIR), our approach enhances restoration in the pixel space. Additionally, we introduce **image-video mixed training** and the **frame-by-frame VAE encoder-decoder** to better adapt to VSR training.
>
> 3. **Experiment Results.** Table 1a in the main paper demonstrates the effectiveness of our training strategy. We further show the results here.
>
>    | Training Stage |  Stage-1  | Stage-1+Stage-2(Image) | Stage-1+Stage-2(Image/Video) |
>    | -------------- | :-------: | :--------------------: | :--------------------------: |
>    | PSNR↑          | **27.20** |         26.39          |            26.48             |
>    | LPIPS ↓        |  0.3037   |         0.2784         |          **0.2696**          |
>    | CLIP-IQA↑      |  0.3236   |         0.5085         |          **0.5107**          |
>    | DOVER↑         |  0.6154   |         0.7694         |          **0.7809**          |
>
> 4. **(Another Novelty) Data Pipeline.** Another contribution is our **video processing pipeline**, through which we construct the **HQ-VSR dataset**. The dataset also contributes to the performance.
>
> In general, while individual loss functions are standard, the overall training strategy, which is adapted for VSR, is novel.
>
>
>
> `Q2-2` Table 1c provides an initial indication of the significance of the proposed HQ-VSR dataset. My question focuses on two key aspects: how much the performance of the proposed model degrades without training or adaptation using HQ-VSR, and to what extent related work or baseline methods might improve if they were evaluated using the same dataset.
>
> `A2-2` Thank you for the insightful question. We conduct two new experiments: **(1)** train our DOVE on the REDS dataset; **(2)** retrain the recent method (MGLD-VSR) with our proposed HQ-VSR dataset.
>
>
>
> **(1) DOVE Trained on REDS.**
>
> We train DOVE using the REDS dataset with the same training settings described in our paper. The stage-2 image dataset is still DIV2K. We refer to the resulting model as **DOVE (REDS)**. The results on UDM10 are shown below:
>
> | Method               | PSNR↑             | LPIPS ↓              | CLIP-IQA↑            | DOVER↑               |
> | -------------------- | ----------------- | -------------------- | -------------------- | -------------------- |
> | MGLD-VSR             | 24.23             | 0.3272               | 0.4557               | 0.7264               |
> | DOVE (REDS)          | 25.28             | 0.3113               | 0.4801               | 0.7510               |
> | DOVE (HQ-VSR, paper) | **26.48** (+1.20) | **0.2696** (−0.0417) | **0.5107** (+0.0306) | **0.7809** (+0.0299) |
>
> These results show that training on HQ-VSR boosts both fidelity and perceptual quality.
>
> Meanwhile, thanks to the proposed training strategy and pretrained model prior, DOVE (REDS) still outperforms existing methods when trained on REDS.
>
>
>
> **(2) MGLD-VSR Trained on HQ-VSR.**
>
> We retrain MGLD-VSR using its official codebase, replacing the original training dataset (REDS) with HQ-VSR.
>
> We use 4 A6000 GPUs with a batch size of 16. Other training settings are consistent with the official configuration.
>
> The resulting model is denoted as **MGLD-VSR (HQ-VSR)**. The performance on UDM10 is:
>
> | Method                    | PSNR↑             | LPIPS ↓              | CLIP-IQA↑            | DOVER↑               |
> | ------------------------- | ----------------- | -------------------- | -------------------- | -------------------- |
> | MGLD-VSR (official, REDS) | 24.23             | 0.3272               | 0.4557               | 0.7264               |
> | MGLD-VSR (HQ-VSR)         | **24.91** (+0.68) | **0.3097** (−0.0175) | **0.4712** (+0.0155) | **0.7383** (+0.0119) |
>
> Training MGLD-VSR on HQ-VSR also results in improvements across all metrics, demonstrating the general utility of our HQ-VSR.
>
>
>
> `Q2-3` L_S2 has three components (MSE, Dists, and frame). How does the model perform if one or two elements are removed?
>
> `A2-3` Thank you for the suggestion. We conduct an ablation study on the loss functions used in stage-2. The results are as follows:
>
> | Method          | PSNR↑ |  LPIPS ↓   | CLIP-IQA↑  |   DOVER↑   |
> | --------------- | :---: | :--------: | :--------: | :--------: |
> | MSE             | 27.50 |   0.3095   |   0.2934   |   0.5455   |
> | MSE+DISTS       | 26.33 |   0.2722   |   0.5044   |   0.7779   |
> | MSE+DISTS+Frame | 26.48 | **0.2696** | **0.5107** | **0.7809** |
>
> We observe that MSE alone has poor perceptual performance, although PSNR is high. This may be because the model tends to output smoother results. The DISTS loss significantly improves **perceptual** quality, as reflected in LPIPS, CLIP-IQA, and DOVER scores. Meanwhile, the Frame loss also brings certain performance improvements by enhancing frame consistency.

---

> > ### Comment · Reviewer_Xx99 · 2025-08-02
> > **Review after rebuttal**
> >
> > Thank you to the authors for the detailed explanation of your work. My concerns have been fully addressed, and I will be raising my vote accordingly

---

> > > ### Author Response · Authors · 2025-08-02
> > > **Thanks Reviewer Xx99 for approving our work**
> > >
> > > Dear Reviewer Xx99,
> > >
> > > Thank you for your response. We are glad that our response addressed your concerns.
> > >
> > > Best regards,
> > >
> > > Authors

---

### Official Review · Reviewer_1Gww · 2025-07-03

**Clarity:** 3
**Significance:** 3
**Originality:** 3
**Rating:** 4
**Confidence:** 5

**Summary:**

This paper proposes DOVE, a one-step diffusion model for real-world video super-resolution (VSR). By fine-tuning a pretrained video diffusion model (CogVideoX) using a two-stage latent–pixel training strategy, and curating a new dataset (HQ-VSR) with a dedicated video processing pipeline, the method achieves competitive perceptual quality with significantly improved inference speed. Experiments on several benchmarks demonstrate strong performance.

**Questions:**

- How does DOVE perform on un-seleted no-reference IQA metrics such as MUSIQ, MANIQA, and Q-ALIGN? Since the current evaluation mainly reports metrics (CLIP-IQA, DOVER, FasterVQA) that were also used in data filtering, it raises concerns about potential bias. To better assess the model’s generalization in perceptual quality, it would be valuable to compare DOVE against other diffusion-based baselines (Upscale-A-Video, MGLD-VSR, VEnhancer, STAR) under these additional NR-IQA metrics. This would help confirm whether DOVE maintains its superiority across diverse perceptual criteria not seen during data construction.

- Is the choice of **t = 399** in one-step inference empirically justified? Providing a comparison with other values could help readers better understand the effect of this hyperparameter and assess whether the choice generalizes across datasets or degradation types.


- Will the authors release HQ-VSR, pretrained models, and training code for reproducibility?

**Ethical Concerns:**

["NO or VERY MINOR ethics concerns only"]

**Final Justification:**

The author addressed my concerns, so I keep the positive score.

**Limitations:**

yes

**Quality:**

3

**Strengths And Weaknesses:**

Strengths:

- Clear motivation: Addresses the critical inefficiency of multi-step diffusion models for VSR.

- Effective training strategy: Latent-to-pixel two-stage fine-tuning is well justified and validated.

- Strong results: Shows superior performance and speed across multiple datasets.

- HQ-VSR dataset: Well-designed pipeline with multi-stage filtering and motion-aware cropping.

Weaknesses:

- Evaluation bias: Key no-reference metrics (CLIP-IQA, FasterVQA) are also used in dataset filtering, raising concerns about selection bias.

- Limited NR-IQA diversity: Lacks evaluation with other no-reference metrics like MUSIQ, MANIQA, Q-ALIGN.

- No subjective quality study: Lacks human evaluation (e.g., MOS) to support perceptual quality claims.
Over-reliance on pretrained prior: Performance gains may stem largely from CogVideoX, not from method design alone.

---

> ### Author Rebuttal · Authors · 2025-07-29
>
> # Response to Reviewer 1Gww (denoted as R1)
>
> `Q1-1` Evaluation bias: Key no-reference metrics (CLIP-IQA, FasterVQA) are also used in dataset filtering, raising concerns about selection bias.
>
> `A1-1` Thank you for the comment.
>
> The metrics in our dataset pipeline are general and diverse, covering both frame-level quality (CLIP-IQA, Aesthetics) and overall video quality (FasterVQA, DOVER). This ensures a comprehensive enhancement of the perceptual performance of the model, rather than overfitting to specific metrics. The results in `A1-2` further support this point.
>
>
>
> `Q1-2` Limited NR-IQA diversity: Lacks evaluation with other no-reference metrics like MUSIQ, MANIQA, Q-ALIGN.
>
> `A1-2` Thank you for the suggestion. We conduct more evaluations and clarify the reasons for only CLIP-IQA.
>
> **(1) Evaluation.**
> We evaluate our method (DOVE), and current SOTA approaches (Upscale-A-Video, MGLD-VSR, VEnhancer, and STAR) on the UDM10 dataset using additional NR-IQA metrics: MUSIQ, MANIQA, and Q-Align. The results are as follows:
>
> | Method          |  MUSIQ↑   |  MANIQA↑   |  Q-Align↑  |
> | --------------- | :-------: | :--------: | :--------: |
> | Upscale-A-Video |   59.06   |   0.2868   |   0.7625   |
> | MGLD-VSR        |   60.55   |   0.2783   |   0.7794   |
> | VEnhancer       |   37.25   |   0.2120   |   0.5513   |
> | STAR            |   41.98   |   0.2088   |   0.5340   |
> | DOVE (ours)     | **61.68** | **0.3284** | **0.8250** |
>
> Our method also achieves strong performance, confirming the absence of bias.
>
>
>
> **(2) Clarification.**
>
> In the paper, we apply a wide range of metrics, including FR-IQA, NR-IQA, and VQA, to ensure comprehensive evaluation. Due to space limitations and the fact that NR-IQA typically measures only single-frame quality rather than full video quality, we primarily report the **representative** NR-IQA (CLIP-IQA) results.
>
> Thanks again for your suggestion. We will include more no-reference metrics in the **revision**.
>
>
>
> `Q1-3` No subjective quality study: Lacks human evaluation (e.g., MOS) to support perceptual quality claims. Over-reliance on pretrained prior: Performance gains may stem largely from CogVideoX, not from method design alone.
>
> `A1-3` Thank you for the comment. We address both concerns by conducting a **user study** and clarifying the role of pretrained **priors**.
>
> **(1) User-Study.**
>
> ***Setup.*** We conduct a user study involving 20 volunteers. DOVE is compared against four SOTA methods: RealBasicVSR, Upscale-A-Video, MGLD-VSR, and STAR. We randomly select 50 videos from three real-world datasets (RealVSR, MVSR4x, and VideoLQ). Each participant is shown 10 randomly selected samples (from the 50 videos). We show the LR input and recovery results of all methods for LR (shown in a randomly anonymous order). Participants are asked to vote for the best quality result for each sample.
>
> In total, we collect **200** votes. The results are as follows:
>
> | Method          | RealBasicVSR | Upscale-A-Video | MGLD-VSR | STAR  | DOVE (ours) |
> | --------------- | :----------: | :-------------: | :------: | :---: | :---------: |
> | Number of votes |      4       |       11        |    32    |  25   |   **128**   |
> | Percentage      |     2.0%     |      5.5%       |  16.0%   | 12.5% |  **64.0%**  |
>
> ***Result.*** DOVE receives **128 votes**, accounting for **64%** of the total, significantly outperforming other methods. These findings align with the quantitative and qualitative results in the paper.
>
>
>
> **(2) Pretrained Prior.**
>
> 1. Leveraging pretrained models is a **feature** of our method and also a corresponding **advantage**. Besides, many recent VSR approaches also build upon pretrained backbones.
> 2. However, to achieve the performance of DOVE, it is not enough to only utilize pretrained models. It depends on our proposed **latent-pixel training strategy** and the tailored **video processing pipeline** (*i.e.*, HQ-VSR). Tables 1a and 1c in the main paper highlight the contributions of these designs.
> 3. Moreover, the non-specific model design feature of our method also ensures **generalizability**. To validate this, we extend our method (training strategy and dataset) to another pretrained T2V model, **WAN2.1-T2V-1.3B** [**ref1**]. The resulting model (denoted as **DOVE (WAN2.1)**), trained with our approach, also performs well. We show results on UDM10:
>
> | Method          |   PSNR↑   |   LPIPS↓   | CLIP-IQA↑  |   DOVER↑   |
> | --------------- | :-------: | :--------: | :--------: | :--------: |
> | Upscale-A-Video |   21.72   |   0.4116   |   0.4697   |   0.7291   |
> | MGLD-VSR        |   24.23   |   0.3272   |   0.4557   |   0.7264   |
> | STAR            |   23.47   |   0.4242   |   0.2417   |   0.4830   |
> | DOVE (WAN2.1)   | **25.28** | **0.2555** | **0.5133** | **0.7563** |
>
> [**ref1**] Wan: Open and Advanced Large-Scale Video Generative Models, 2025.
>
>
>
> `Q1-4` How does DOVE perform on un-seleted no-reference IQA metrics such as MUSIQ, MANIQA, and Q-ALIGN? Since the current evaluation mainly reports metrics (CLIP-IQA, DOVER, FasterVQA) that were also used in data filtering, it raises concerns about potential bias. To better assess the model’s generalization in perceptual quality, it would be valuable to compare DOVE against other diffusion-based baselines (Upscale-A-Video, MGLD-VSR, VEnhancer, STAR) under these additional NR-IQA metrics. This would help confirm whether DOVE maintains its superiority across diverse perceptual criteria not seen during data construction.
>
> `A1-4` Thank you for the comment.
>
> The metrics used in data filtering are general and diverse, which minimizes the risk of selection bias. We provide additional evaluations using other NR-IQA metrics: MUSIQ, MANIQA, and Q-Align.
>
> For details, refer to `A1-1` and `A1-2`.
>
>
>
> `Q1-5` Is the choice of t = 399 in one-step inference empirically justified? Providing a comparison with other values could help readers better understand the effect of this hyperparameter and assess whether the choice generalizes across datasets or degradation types.
>
> `A1-5` Thank you for the suggestion.
>
> **(1) Ablation on Different Values of $t$.** We conduct an ablation study on the denoising step *t*, as reported in Tab. 3 of the **supplementary material**. We show part of the primary results below:
>
> | Step (t)  |   99   |    399     |    599    |
> | --------- | :----: | :--------: | :-------: |
> | PSNR↑     | 27.22  |   27.20    | **27.28** |
> | LPIPS↓    | 0.3115 | **0.3037** |  0.3138   |
> | CLIP-IQA↑ | 0.3116 | **0.3236** |  0.3036   |
> | DOVER↑    | 0.5862 | **0.6154** |  0.5543   |
>
> Setting $t=399$ achieves a better balance between fidelity and perceptual quality. These results align with our analysis in Sec. 3.1 in the main paper. More detailed results and analyses are in the supplementary material.
>
> **(2) Generalization across Datasets (REDS).**  We further conduct the same experiment on another training dataset (**REDS**), with other settings remaining unchanged. The results are as follows:
>
> | Step (t)  |    99     |    399     |  599   |
> | --------- | :-------: | :--------: | :----: |
> | PSNR↑     | **26.45** |   26.35    | 26.28  |
> | LPIPS↓    |  0.3478   | **0.3378** | 0.3435 |
> | CLIP-IQA↑ |  0.3747   | **0.3853** | 0.3813 |
> | DOVER↑    |  0.6435   | **0.6611** | 0.6331 |
>
> The results are close to those trained on HQ-VSR, which shows that the choice of $t$ is generalizable.
>
>
>
> `Q1-6` Will the authors release HQ-VSR, pretrained models, and training code for reproducibility?
>
> `A1-6` Yes, we will release the code, HQ-VSR dataset, and pretrained models to support full reproducibility.

---

> > ### Comment · Reviewer_1Gww · 2025-08-06
> >
> > Thank you for the detailed responses and the additional experiments, which greatly strengthen the paper.
> >
> > As a minor suggestion, since most evaluations are conducted on synthetic datasets with predefined degradations, results may benefit from training-testing consistency. To better assess real-world performance, it would be helpful to include no-reference IQA comparisons (e.g., Q-ALIGN, MUSIQ, MANIQA) on a real-world dataset like VideoLQ, which contains only real LQ inputs without GT. This would offer a more comprehensive view of perceptual quality in practical scenarios.

---

> ### Author Response · Authors · 2025-08-06
> **Further discussion with Reviewer 1Gww**
>
> Dear Review 1Gww,
>
> Thank you for taking the time to review our paper and for your thoughtful and constructive comments. We have carefully considered your comments and provided detailed responses to your questions:
>
> 1. We clarify our selection of NR-IQA metrics and conduct evaluations using more no-reference metrics to demonstrate the perceptual generalization ability of our method.
> 2. We perform a user study to support the perceptual quality. We also provide a more detailed explanation of using pretrained diffusion priors in our method.
> 3. We conduct an ablation study on the timestep t to justify our choice.
>
> As the author–reviewer discussion phase is drawing to a close, we would like to confirm whether our responses have addressed your concerns. If you have any questions or suggestions, please let us know. We would be grateful to hear them.
>
> Thank you again for your time, effort, and valuable feedback.
>
>
>
> Best regards,
>
> Authors

---

> ### Author Response · Authors · 2025-08-06
> **Further Discussion on Real-World Dataset**
>
> Dear Review 1Gww,
>
> Thank you for your response. First, we are pleased that you appreciated our response and experiment.
>
> Second, regarding your suggestion to include NR-IQA evaluation on the real-world dataset, we fully agree that this is both meaningful and valuable. We provide results on the VideoLQ dataset below:
>
> | Method      | MUSIQ↑ | MANIQA↑ | Q-Align↑ |
> | ----------- | :----: | :-----: | :------: |
> | MGLD-VSR    | 46.20  | 0.2711  |  0.7882  |
> | VEnhancer   | 42.35  | 0.2764  |  0.7489  |
> | STAR        | 39.66  | 0.2519  |  0.7761  |
> | DOVE (ours) | 48.72  | 0.2802  |  0.8038  |
>
> Our method performs well across diverse NR-IQA metrics, further supporting its generalization. We will incorporate these results in the revision of the paper to provide a more comprehensive evaluation.
>
> Thank you again for your thoughtful feedback and helpful suggestions.
>
> Best regards,
>
> Authors

---

> > ### Comment · Reviewer_1Gww · 2025-08-08
> >
> > Thank you for your detailed response and for providing the NR-IQA results on the real-world dataset, which indeed strengthen the generalization claims of your method.
> >
> > I would also like to raise a broader question regarding the current development trend in the VSR field. Both ISR and VSR seem increasingly reliant on powerful pretrained image (such as FLUX and Qwen-Image) or video generation models (Wan 2.2). While fine-tuning such models for downstream tasks like VSR often yields strong performance, one might argue that much of the gain comes from leveraging the generative capacity of these large models, with fine-tuning serving primarily as a domain adaptation step.
> >
> > I’m curious to hear your thoughts on this perspective. Do you see this reliance as a limitation or an opportunity? And more broadly, how do you envision the future direction of ISR/VSR research?

---

> > > ### Author Response · Authors · 2025-08-09
> > > **Further Discussion on Future Direction**
> > >
> > > Dear Review 1Gww,
> > >
> > > Thank you for your comments. We are pleased that our responses have addressed your questions regarding generalization on real-world datasets.
> > >
> > > Regarding your discussion on the reliance on large pretrained models and future direction, our perspective is as follows:
> > >
> > > 1. While methods built on pretrained models benefit from large models' strong generative capacity, the fine-tuning process and task-specific design ultimately determine the upper bound of performance. In SR, it is not only realism that matters but also fidelity. Thus, performance cannot be attributed solely to the generative prior. This is even more evident in image SR, where many approaches apply the same model (*i.e.*, Stable Diffusion), yet achieve markedly different results.
> > > 2. We think using large pretrained models is both an opportunity and a likely future trend. Because visual tasks are inherently interconnected, better visual understanding leads to better visual processing results. The continuous development of large pre-trained models is constantly strengthening this. This parallels the post-GPT era in NLP, where large language models became mainstream for various downstream tasks. We expect a similar trajectory in the vision domain.
> > > 3. For future direction, we think ISR/VSR will evolve toward being more intelligent, realistic, and generalizable. That is, the model can adapt to diverse scenarios, enable user control, and handle cross-task processing. This requires adapting pretrained models to different tasks, and ensuring that the models remain flexible and controllable.
> > >
> > > Thank you again for your insightful comments. We welcome further discussion.
> > >
> > > Best regards,
> > >
> > > Authors

---

### Author Response · Authors · 2025-08-09
**Response to all reviewers and area chairs for a brief summary**

Dear reviewers and area chairs,

We sincerely thank all reviewers and area chairs for their valuable time and thoughtful comments. After several rounds of discussions with the reviewers, we would like to provide a summary.

**All reviewers** (R1–1Gww, R2–Xx99, R3–WLko, R4–svnJ) confirm that our responses address their concerns and approve of our work.



We are pleased that:

1. All reviewers recognize the efficiency and impressive performance of our one-step method.
2. R1 and R4 find our latent-space training strategy effective.
3. R1 and R3 acknowledge the value of our video processing pipeline.

We have provided detailed responses to each reviewer and would like to offer a summary:

1. We evaluate on more **NR-IQA** metrics to demonstrate generalization.
2. We conduct a **user study** to validate perceptual quality.
3. We clarify the role and usage of pretrained diffusion priors.
4. We perform more **ablation studies**, including timestep, loss function, image ratio, and data filtering components.
5. We clarify the **novelty** of our method and its **distinctions** from existing approaches.
6. We demonstrate the importance of **HQ-VSR** through experiments.
7. We extend our approach to **another pretrained model** (WAN2.1) to show generalizability.
8. We provide an interpretation of our DOVE as a **diffusion model**.
9. We compare **more advanced methods**, including diffusion-based SR, GAN-based VSR, and the one-step method.
10. We analyze the problems with **visual results**.



We again thank all reviewers and area chairs.

Best regards,

Authors

---

### Decision · Program_Chairs · 2025-09-17

**Decision:**

Accept (poster)

**Comment:**

This paper introduces DOVE, a one-step diffusion model for video super-resolution. The motivation is clear: multi-step diffusion methods are too slow for real-world deployment, and the proposed approach achieves much faster inference while maintaining strong perceptual quality. The main strengths are the effective latent-to-pixel fine-tuning strategy, strong benchmark results, and the introduction of the HQ-VSR dataset. Concerns were raised about the performance gains being largely attributable to the pretrained base model, limited technical novelty, and possible evaluation bias. During rebuttal, the authors provided additional comparisons with other SOTA methods using more diverse metrics and included human studies, which addressed most concerns and strengthened their claims. Reviewers converged towards acceptance after these clarifications. Overall, despite modest novelty, the paper’s contributions are solid and well-supported after rebuttal, and I recommend acceptance as a poster.